# META-GMVAE: MIXTURE OF GAUSSIAN VAES FOR UNSUPERVISED META-LEARNING

**Dong Bok Lee[1], Dongchan Min[1], Seanie Lee[1], and Sung Ju Hwang[1,2]**
KAIST[1], AITRICS[2], South Korea
{markhi,alsehdcks95,lsnfamily02,sjhwang82}@kaist.ac.kr

## ABSTRACT

Unsupervised learning aims to learn meaningful representations from unlabeled data which can capture its intrinsic structure, that can be transferred to downstream tasks. Meta-learning, whose objective is to learn to generalize across tasks such that the learned model can rapidly adapt to a novel task, shares the spirit of unsupervised learning in that the both seek to learn more effective and efficient learning procedure than learning from scratch. The fundamental difference of the two is that the most meta-learning approaches are supervised, assuming full access to the labels. However, acquiring labeled dataset for meta-training not only is costly as it requires human efforts in labeling but also limits its applications to pre-defined task distributions. In this paper, we propose a principled unsupervised meta-learning model, namely Meta-GMVAE, based on Variational Autoencoder (VAE) and set-level variational inference. Moreover, we introduce a mixture of Gaussian (GMM) prior, assuming that each modality represents each class-concept in a randomly sampled episode, which we optimize with Expectation-Maximization (EM). Then, the learned model can be used for downstream few-shot classification tasks, where we obtain task-specific parameters by performing semi-supervised EM on the latent representations of the support and query set, and predict labels of the query set by computing aggregated posteriors. We validate our model on Omniglot and Mini-ImageNet datasets by evaluating its performance on downstream few-shot classification tasks. The results show that our model obtains impressive performance gains over existing unsupervised meta-learning baselines, even outperforming supervised MAML on a certain setting.

## 1 INTRODUCTION

*Unsupervised learning* is one of the most fundamental and challenging problems in machine learning, due to the absence of target labels to guide the learning process. Thanks to the enormous research efforts, there now exist many unsupervised learning methods that have shown promising results on real-world domains, including image recognition (Le, 2013) and natural language understanding (Ramachandran et al., 2017). The essential goal of unsupervised learning is obtaining meaningful feature representations that best characterize the data, which can be later utilized to improve the performance of the downstream tasks, by training a supervised task-specific model on the top of the learned representations (Reed et al., 2014; Cheung et al., 2015; Chen et al., 2016) or fine-tuning the entire pre-trained models (Erhan et al., 2010).

*Meta-learning*, whose objective is to learn general knowledge across diverse tasks, such that the learned model can rapidly adapt to novel tasks, shares the spirit of unsupervised learning in that both seek more efficient and effective learning procedure over learning from scratch. However, the essential difference between the two is that most meta-learning approaches have been built on the supervised learning scheme, and require human-crafted task distributions to be applied in few-shot classification. Acquiring labeled dataset for meta-training may require a massive amount of human efforts, and more importantly, meta-learning limits its applications to the pre-defined task distributions (e.g. classification of specific set of classes).

Two recent works have proposed *unsupervised meta-learning* that can bridge the gap between unsupervised learning and meta-learning by focusing on constructing supervised tasks with pseudo-labels from the unlabeled data. To do so, CACTUs (Hsu et al., 2019) clusters data in the embedding space

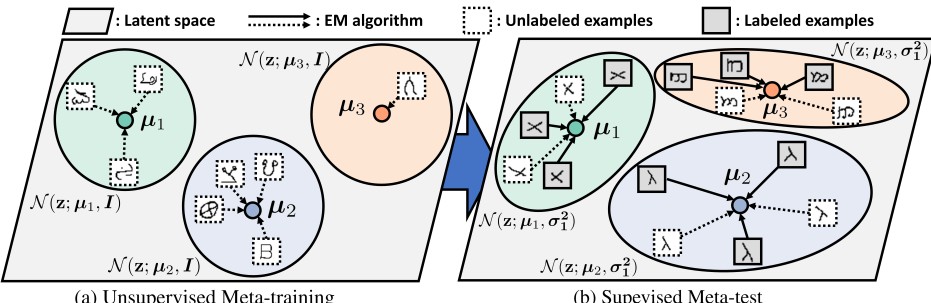

Figure 1: During meta-training, Meta-GMVAE learns multi-modal latent space that can best explain the unlabeled data using EM algorithm. At meta-test time, we use semi-supervised EM to map both the support (labeled data) and queries (unlabeled data) to each mode learned during meta-training.

learned with several unsupervised learning methods, while UMTRA (Khodadadeh et al., 2019) assumed that each randomly drawn sample represents a different class and augmented each pseudo-class with data augmentation (Cubuk et al., 2018). After constructing the meta-training dataset with such heuristics, they simply apply supervised meta-learning algorithms as usual. Despite the success of the existing unsupervised meta-learning methods, they are fundamentally limited, since 1) they only consider unsupervised learning for heuristic pseudo-labeling of unlabeled data, and 2) the two-stage approach makes it impossible to recover from incorrect pseudo-class assignment when learning the unsupervised representation space.

In this paper, we propose a principled unsupervised meta-learning model based on Variational Autoencoder (VAE) (Kingma & Welling, 2014) and set-level variational inference using self-attention (Vaswani et al., 2017). Moreover, we introduce multi-modal prior distributions, a mixture of Gaussians (GMM), assuming that each modality represents each class-concept in any given tasks. Then the parameter of GMM is optimized by running *Expectation-Maximization* (EM) on the observations sampled from the set-dependent variational posterior. In this framework, however, there is no guarantee that each modality obtained from EM algorithm corresponds to a label. To realize modality as label, we deploy semi-supervised EM at meta-test time, considering the support set and query set as labeled and unlabeled observations, respectively. We refer to our method as *Meta-Gaussian Mixture Variational Autoencoders* (Meta-GMVAE) (See Figure 1 for high-level concept). While our method can be used as a full generative model for generating the samples (images), the ability to generalize to generate samples may not be necessary for capturing the meta-knowledge for non-generative downstream tasks. Thus, we propose another version of Meta-GMVAE that reconstructs high-level features learned by unsupervised representation learning approaches (e.g. Chen et al. (2020)).

To investigate the effectiveness of our framework, we run experiments on two benchmark few-shot image classification datasets, namely Omiglot (Lake et al., 2011) and Mini-Imagenet (Ravi & Larochelle, 2017). The experimental results show that our Meta-GMVAE obtains impressive performance gains over the relevant unsupervised meta-learning baselines on both datasets, obtaining even better accuracy than fully supervised MAML (Finn et al., 2017) while utilizing as small as $0.1\%$ of the labeled data on one-shot settings in Omniglot dataset. Moreover, our model can generalize to classification tasks with different number of ways (classes) without loss of accuracy. Our contribution is threefold:

- We propose a novel unsupervised meta-learning model, namely Meta-GMVAE, which meta-learns the set-conditioned prior and posterior network for a VAE. Our Meta-GMVAE is a principled unsupervised meta-learning method, unlike existing methods on unsupervised meta-learning that combines heuristic pseudo-labeling with supervised meta-learning.

- We propose to learn the multi-modal structure of a given dataset with the Gaussian mixture prior, such that it can adapt to a novel dataset via the EM algorithm. This flexible adaptation to a new task, is not possible with existing methods that propose VAEs with Gaussian mixture priors for single task learning.

- We show that Meta-GMVAE largely outperforms relevant unsupervised meta-learning baselines on two benchmark datasets, while obtaining even better performance than a supervised meta-learning model under a specific setting. We further show that Meta-GMVAE can generalize to classification tasks with different number of ways (classes).

## 2 RELATED WORK

**Unsupervised learning** Many prior unsupervised learning methods have developed proxy objectives which is either based on reconstruction (Vincent et al., 2010; Higgins et al., 2017), adversarially obtained image fidelity (Radford et al., 2016; Salimans et al., 2016; Donahue et al., 2017; Dumoulin et al., 2017), disentanglement (Bengio et al., 2013; Reed et al., 2014; Cheung et al., 2015; Chen et al., 2016; Mathieu et al., 2016; Denton & Birodkar, 2017; Kim & Mnih, 2018; Ding et al., 2020), clustering (Coates & Ng, 2012; Krähenbühl et al., 2016; Bojanowski & Joulin, 2017; Caron et al., 2018), or contrastive learning (Chen et al., 2020). In the unsupervised learning literature, the most relevant work to ours are methods that use Gaussian Mixture priors for variational autoencoders. Dilokthanakul et al. (2016); Jiang et al. (2017) consider single task learning and therefore, the learned prior parameter is fixed after training, and thus cannot adapt to new tasks. CURL (Rao et al., 2019) learns a network that outputs Gaussian mixture priors over a sequence of tasks for unsupervised continual learning. However CURL cannot adapt to a new task without training on it, while our framework can generalize to a new task without any training, via amortized inference with a dataset (task) encoder. Also, our model does not learn Gaussian mixture priors but rather obtain them on the fly using the expectation-maximization algorithm.

**Meta-learning** Meta-learning (Thrun & Pratt, 1998) shares the intuition of unsupervised learning in that it aims to improve the model performance on an unseen task by leveraging prior knowledge, rather than learning from scratch. While the literature on meta-learning is vast, we only discuss relevant existing works for few-shot image classification. Metric-based meta-learning (Koch et al., 2015; Vinyals et al., 2016; Snell et al., 2017; Oreshkin et al., 2018; Mishra et al., 2018) is one of the most popular approaches, where it learns to embed the data instances of the same class to be closer in the shared embedding space. One can measure the distance in the embedding space by cosine similarity (Vinyals et al., 2016), or Euclidean distance (Snell et al., 2017). On the other hand, gradient-based meta-learning (Finn et al., 2017; 2018; Li et al., 2017; Lee & Choi, 2018; Ravi & Beatson, 2019; Flennerhag et al., 2020) aims at learning a global initialization of parameters, which can rapidly adapt to a novel task with only a few gradient steps. Moreover, some previous works (Hewitt et al., 2018; Edwards & Storkey, 2017; Garnelo et al., 2018) tackle meta-learning by modeling the set-dependent variational posterior with a single global latent variable, however, we model the variational posterior conditioned on each data instances. Moreover, while all of these works assume supervised learning scenarios where one has access to full labels in meta-training stage, we focus on unsupervised setting in this paper.

**Unsupervised meta-learning** One of the main limitations of conventional meta-learning methods is that their application is strictly limited to the tasks from a pre-defined task distribution. A few works (Hsu et al., 2019; Khodadadeh et al., 2019) have been proposed to resolve this issue by combining unsupervised learning with meta-learning. The main idea is to construct meta-training dataset in an unsupervised manner by leveraging existing supervised meta-learning models. CACTUs (Hsu et al., 2019) deploy several deep metric learning (Berthelot et al., 2019; Donahue et al., 2017; Caron et al., 2018; Chen et al., 2016) to episodically cluster the unlabeled dataset, and then train MAML (Finn et al., 2017) and Prototypical Networks (Snell et al., 2017) on the constructed data. UMTRA (Khodadadeh et al., 2019) assumes that each randomly drawn sample is from a different class from others, and use data augmentation (Cubuk et al., 2018) to construct synthetic task distribution for meta-training. Instead of only deploying unsupervised learning for constructing meta-training task distributions, we propose an unsupervised meta-learning model that meta-learns set-level variational posterior by matching the multi-modal prior distribution representing latent classes.

## 3 UNSUPERVISED META-LEARNING WITH META-GMVAEs

In this section, we describe our problem setting with respect to unsupervised meta-learning, and demonstrate our approach. The graphical illustration of our model for unsupervised meta-training and supervised meta-test is depicted in Figure 2.

### 3.1 PROBLEM STATEMENT

Our goal is to learn unsupervised feature representations which can be transferred to wide range of downstream few-shot classification tasks. As suggested by Hsu et al. (2019); Khodadadeh et al.

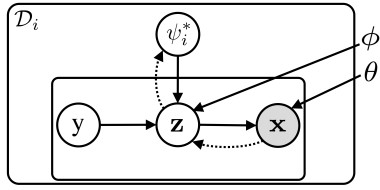
(a) Unsupervised Meta-training

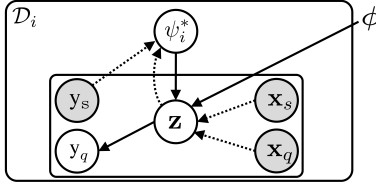
(b) Supevised Meta-test

Figure 2: The graphical illustration of Meta-GMVAE. The dotted lines denote either variational inference or Expectation Maximization. (a): We introduce the multimodal distribution $p_\psi(\mathbf{z})$ into prior distribution, and its optimal task-specific parameter $\psi_i^*$ is obtained by EM in an episodic manner. (b): For meta-test, we obtain task-specific parameter $\psi_i^*$ by semi-supervised EM using $\mathbf{x}_s$, $\mathbf{y}_s$, and $\mathbf{x}_q$.

(2019), we only assume an unlabeled dataset $\mathcal{D}_u = \{\mathbf{x}_u\}_{u=1}^U$ in the meta-training stage. We aim toward applying the knowledge learned during unsupervised meta-training stage to novel tasks in meta-test stage, which comes with a modest amount of labeled data (or as few as a single example per class) for each task. As with most meta-learning methods, we further assume that the labeled data are drawn from the same distribution as that of the unlabeled data, with a different set of classes. Specifically, the goal of a $K$-way $S$-shot classification task $\mathcal{T}$ is to correctly predict the labels of query data points $\mathcal{Q} = \{\mathbf{x}_q\}_{q=1}^Q$, using $S$ support data points and labels $\mathcal{S} = \{(\mathbf{x}_s, \mathbf{y}_s)\}_{s=1}^S$ per class, where $S$ is relatively small (i.e. between 1 and 50).

## 3.2 META-LEVEL GAUSSIAN MIXTURE VAE

**Unsupervised meta-training** We now describe the meta-learning framework for learning unsupervised latent representations that can be transferred to human-designed few-shot image-classification tasks. In particular, we aim toward learning multi-modal latent spaces for Variational Autoencoder (VAE) in an episodic manner. We use the Gaussian mixture for the prior distribution $p_\psi(\mathbf{z}) = \sum_{k=1}^K p_\psi(\mathbf{y} = k)p_\psi(\mathbf{z}|\mathbf{y} = k)$, where $\psi$ is the parameter of the prior network. Then the generative process can be described as follows:

- $\mathbf{y} \sim p_\psi(\mathbf{y})$, where y corresponds to the categorical L.V. for a single mode.
- $\mathbf{z} \sim p_\psi(\mathbf{z}|\mathbf{y})$, where z corresponds to the Gaussian L.V. responsible for data generation.
- $\mathbf{x} \sim p_\theta(\mathbf{x}|\mathbf{z})$, where $\theta$ is the parameter of the generative model.

The above generative process is similar to those from the previous works (Dilokthanakul et al., 2016; Jiang et al., 2017) on modeling the VAE prior with Gaussian mixtures. However, they target single-task learning and the parameter of the prior network is fixed after training such as equation 1c in Dilokthanakul et al. (2016) and equation 5 in Jiang et al. (2017), which is suboptimal since a meta-learning model should be able to adapt and generalize to a novel task.

To learn the set-dependent multi-modalities, we further assume that there exists a parameter $\psi_i$ for each episodic dataset $\mathcal{D}_i = \{\mathbf{x}_j\}_{j=1}^M$, which is randomly drawn from the unlabeled dataset $\mathcal{D}_u$. Then we derive the variational lower bound for the marginal log-likelihood of $\mathcal{D}_i$ as follows:

$$\log p_\theta(\mathcal{D}_i) = \sum_{j=1}^M \log p_\theta(\mathbf{x}_j) = \sum_{j=1}^M \log \int p_\theta(\mathbf{x}_j|\mathbf{z}_j)p_{\psi_i}(\mathbf{z}_j)\frac{q_\phi(\mathbf{z}_j|\mathbf{x}_j, \mathcal{D}_i)}{q_\phi(\mathbf{z}_j|\mathbf{x}_j, \mathcal{D}_i)}d\mathbf{z}_j \tag{1}$$

$$\geq \sum_{j=1}^M \left[ \mathbb{E}_{\mathbf{z}_j \sim q_\phi(\mathbf{z}_j|\mathbf{x}_j, \mathcal{D}_i)} \left[ \log p_\theta(\mathbf{x}_j|\mathbf{z}_j) + \log p_{\psi_i}(\mathbf{z}_j) - \log q_\phi(\mathbf{z}_j|\mathbf{x}_j, \mathcal{D}_i)) \right] \right] \tag{2}$$

$$\approx \sum_{j=1}^M \frac{1}{N} \sum_{n=1}^N \left[ \log p_\theta(\mathbf{x}_j|\mathbf{z}_j^{(n)}) + \log p_{\psi_i}(\mathbf{z}_j^{(n)}) - \log q_\phi(\mathbf{z}_j^{(n)}|\mathbf{x}_j, \mathcal{D}_i) \right] \tag{3}$$

$$=: \mathcal{L}(\theta, \phi, \psi_i, \mathcal{D}_i), \quad \mathbf{z}_j^{(n)} \overset{i.i.d}{\sim} q_\phi(\mathbf{z}_j|\mathbf{x}_j, \mathcal{D}_i). \tag{4}$$

Here the lower bound for each datapoint is approximated by Monte Carlo estimation with the sample size $N$. Following the convention of the VAE literature, we assume that the variational posterior $q_\phi(\mathbf{z}_j|\mathbf{x}_j, \mathcal{D}_i)$ follows an isotropic Gaussian distribution.

| **Algorithm 1** Meta-training | **Algorithm 2** Meta-test for an episode |
|---|---|
| **Require:** An unlabeled dataset $\mathcal{D}_u$ | **Require:** A test task $\mathcal{T} = \mathcal{S} \cup \mathcal{Q}$ |
| 1: Initialize parameters $\theta, \phi$ | 1: Set $\mathcal{D} = \{\mathbf{x}_s\}_{s=1}^{S} \cup \{\mathbf{x}_q\}_{q=1}^{Q}$ |
| 2: **while** not done **do** | 2: Draw $n$ MC samples from $q_\phi(\mathbf{z}_j|\mathbf{x}_j, \mathcal{D})$ |
| 3:     Sample $B$ episode datasets $\{\mathcal{D}_i\}_{i=1}^{B}$ from $\mathcal{D}_u$ | 3: Initialize $\boldsymbol{\mu}_k = \frac{\sum_{s,n=1}^{S,N} \mathbf{1}_{y_s^{(n)}=k} \mathbf{z}_s^{(n)}}{\sum_{s,n=1}^{S,N} \mathbf{1}_{y_s^{(n)}=k}}$ and $\boldsymbol{\sigma}_k^2 = \boldsymbol{I}$ |
| 4:     **for all** $i \in [1, B]$ **do** | |
| 5:         Draw $n$ MC samples from $q_\phi(\mathbf{z}_j|\mathbf{x}_j, \mathcal{D}_i)$ | |
| 6:         Initialize $\pi_k$ as $1/K$ and randomly choose $K$ different points for $\boldsymbol{\mu}_k$. | 4: Compute optimal parameter $\psi^*$ using Eq 10 |
| 7:         Compute optimal parameter $\psi_i^*$ using Eq 7 | 5: Compute $p(\mathbf{y}_q|\mathbf{x}_q, \mathcal{D})$ using Eq 11 |
| 8:     **end for** | 6: Infer the label $\mathbf{y}_q = \arg\max_k p(\mathbf{y}_q = k|\mathbf{x}_q, \mathcal{D})$ |
| 9:     Update $\theta, \phi$ using $\mathcal{L}(\theta, \phi, \{\mathcal{D}_i\}_{i=1}^{B})$ in Eq 9. | 7: |
| 10: **end while** | 8: |

**Set-dependent variational posterior** Our derivation of the evidence lower bound in Eq 4 is similar to that of the hierarchical VAE framework, such as equation 3 in Edwards & Storkey (2017) and equation 4 in Hewitt et al. (2018), in that we use the i.i.d assumption that the log likelihood of a dataset equals the sum over the log-likelihoods of each individual data point. Yet, previous works assume that each input set consists of data instances from a single concept (e.g. a class), therefore, they encode the dataset into a single global latent variable (e.g. $q_\phi(\mathbf{z}|\mathcal{D})$). This is not appropriate for unsupervised meta-learning where labels are unavailable. Thus we learn a set-conditioned variational posterior $q_\phi(\mathbf{z}_j|\mathbf{x}_j, \mathcal{D}_i)$, which models a latent variable to encode each data $\mathbf{x}_j$ within the given dataset $\mathcal{D}_i$ into the latent space. Specifically, we model the variational posterior $q_\phi(\mathbf{z}_j|\mathbf{x}_j, \mathcal{D}_i)$ using the self-attention mechanism (Vaswani et al., 2017) as follows:

$$H = \text{TransformerEncoder}(f(\mathcal{D}_i))$$
$$\boldsymbol{\mu}_j = W_{\boldsymbol{\mu}} H_j + \mathbf{b}_{\boldsymbol{\mu}}, \quad \boldsymbol{\sigma}_j^2 = exp(W_{\boldsymbol{\sigma^2}} H_j + \mathbf{b}_{\boldsymbol{\sigma^2}}) \tag{5}$$
$$q_\phi(\mathbf{z}_j|\mathbf{x}_j, \mathcal{D}_i) = \mathcal{N}(\mathbf{z}_j; \boldsymbol{\mu}_j, \boldsymbol{\sigma}_j^2)$$

Here we deploy TransformerEncoder($\cdot$), a neural network based on the multi-head self-attention mechanism proposed by Vaswani et al. (2017), to model the dependency between data instances, and $f$ is a convolutional neural network (or an identity function for the Mini-ImageNet) which takes each data in $\mathcal{D}_i$ as an input. Moreover, we use the reparameterization trick (Kingma & Welling, 2014) to train the model with backpropagation since the stochastic sampling process $\mathbf{z}_j^{(n)} \overset{i.i.d}{\sim} q_\phi(\mathbf{z}_j|\mathbf{x}_j, \mathcal{D}_i)$ is non-differentiable.

**Expectation Maximization** As discussed before, we assume that the parameter $\psi_i$ of the prior Gaussian Mixture is task-specific and characterizes the given dataset $\mathcal{D}_i$. To obtain the task-specific parameter that optimally explain the given dataset, we propose to locally maximize the lower bound in Eq 4 with respect to the prior parameter $\psi_i$. We can obtain the optimal parameter $\psi_i^*$ by solving the following optimization problem:

$$\psi_i^* = \arg\max_{\psi_i} \mathcal{L}(\theta, \phi, \psi_i, \mathcal{D}_i) = \arg\max_{\psi_i} \sum_{j,n=1}^{M,N} \log p_\psi(\mathbf{z}_j^{(n)}), \quad \mathbf{z}_j^{(n)} \overset{i.i.d}{\sim} q_\phi(\mathbf{z}_j|\mathbf{x}_j, \mathcal{D}_i), \quad (6)$$

where we only consider the term related to the task-specific parameter $\psi_i$, and eliminate the normalization term $\frac{1}{N}$ since it does not change the solution of the optimization problem. The above formula implies that the optimal parameter maximizes the log-likelihood of observations which can be drawn from the variational posterior distribution. However, we do not have an analytic solution for Maximum Likelihood Estimation (MLE) of a GMM.

The most prevalent approach for estimating the parameters for the mixture of Gaussian is solving it with Expectation Maximization (EM) algorithm. To this end, we propose to optimize the task-specific parameter of GMM prior distribution using EM algorithm as follows:

$$(\text{E-step}) \quad Q_{j,n}(k) := p(\mathbf{y}_j^{(n)} = k|\mathbf{z}_j^{(n)}) = \frac{\pi_k \mathcal{N}(\mathbf{z}_j^{(n)}; \boldsymbol{\mu}_k, \boldsymbol{I})}{\sum_k \pi_k \mathcal{N}(\mathbf{z}_j^{(n)}; \boldsymbol{\mu}_k, \boldsymbol{I})}$$

$$(\text{M-step}) \quad \boldsymbol{\mu}_k := \frac{\sum_{j,n=1}^{M,N} Q_{j,n}(k)\mathbf{z}_j^{(n)}}{\sum_{j,n=1}^{M,N} Q_{j,n}(k)}, \quad \pi_k := \frac{\sum_{j,n=1}^{M,N} Q_{j,n}(k)}{\sum_{k=1}^{K} \sum_{j,n=1}^{M,N} Q_{j,n}(k)} \tag{7}$$

$$\psi_i := \{(\boldsymbol{\mu}_k, \boldsymbol{I}, \pi_k)\}_{k=1}^{K},$$

where $\pi_k$, $\boldsymbol{\mu}_k$, and $\mathcal{N}(\cdot)$ denote the mixing probability of $k$-th component, mean parameter, and normal distribution, respectively. We assume that the covariance matrix of Gaussian distribution is fixed with the identity matrix $\boldsymbol{I}$, following the assumption of original VAE on the prior distribution. We initialize $\{\pi_k\}_{k=1}^{K}$ and $\{\boldsymbol{\mu}_k\}_{k=1}^{K}$ as $\frac{1}{K}$ and randomly drawn $K$ different points, respectively. We can obtain MLE solution for the parameters of GMM, by iteratively performing E-step and M-step until the log-likelihood converges. We found that using a fixed number of iterations for the EM algorithm does not degrade the performance, and consider it as a hyperparameter of our framework.

**Training objective** Note that we want to maximize the variational lower bound of the marginal log-likelihood over all the episode datasets $\mathcal{D}_i$ that can be sampled from $\mathcal{D}_u$. We use stochastic gradient ascent with respect to the variational parameter $\phi$ and the generative parameter $\theta$, to maximize the following objective:

$$\mathcal{L}(\theta, \phi, \{\mathcal{D}_i\}_{i=1}^{B}) := \frac{1}{B} \sum_{i=1}^{B} \left[ \max_{\psi_i} \mathcal{L}(\theta, \phi, \psi_i, \mathcal{D}_i) \right] \tag{8}$$

$$= \frac{1}{B} \sum_{i=1}^{B} \sum_{j=1}^{M} \frac{1}{N} \sum_{n=1}^{N} \left[ \log p_\theta(\mathbf{x}_j | \mathbf{z}_j^{(n)}) + \log p_{\psi_i^*}(\mathbf{z}_j^{(n)}) - \log q_\phi(\mathbf{z}_j^{(n)} | \mathbf{x}_j, \mathcal{D}_i) \right]. \tag{9}$$

Here we use $B$ mini-batch of episode datasets, where each dataset consists of $M$ datapoints. The task-specific parameter $\psi_i^*$ for each episode dataset $\mathcal{D}_i$ is obtained by EM algorithm in Eq 7.

**Supervised meta-test** By introducing the multi-modal prior distribution into a generative learning framework, our model learns pseudo-class concepts by clustering latent features with EM algorithm. However, there is no guarantee that each modality obtained by EM algorithm corresponds to the label we are interested in at the meta-test stage. To realize modality as label in downstream few-shot image classification tasks, we deploy semi-supervised EM algorithm instead. Given a task $\mathcal{T}$ consisting of support set $\mathcal{S} = \{(\mathbf{x}_s, \mathbf{y}_s)\}_{s=1}^{S}$ and query set $\mathcal{Q} = \{\mathbf{x}_q\}_{q=1}^{Q}$, we use both the support set and query set as an episode dataset $\mathcal{D} = \{\mathbf{x}_s\}_{s=1}^{S} \cup \{\mathbf{x}_q\}_{q=1}^{Q}$ and draw latent variables from the variational posterior $q_\phi(\mathbf{z}_j | \mathbf{x}_j, \mathcal{D})$. Note that we abbreviate the index $i$ since we consider a single task for now. We then perform semi-supervised EM algorithm as follows:

$$\text{(E-step)} \quad Q_{q,n}(k) := p(\mathbf{y}_q^{(n)} = k | \mathbf{z}_q^{(n)}) = \frac{\mathcal{N}(\mathbf{z}_q^{(n)}; \boldsymbol{\mu}_k, \boldsymbol{\sigma}_k^2)}{\sum_k \mathcal{N}(\mathbf{z}_q^{(n)}; \boldsymbol{\mu}_k, \boldsymbol{\sigma}_k^2)}$$

$$\text{(M-step)} \quad \begin{aligned} \boldsymbol{\mu}_k &:= \frac{\sum_{s,n=1}^{S,N} \mathbf{1}_{\mathbf{y}_s^{(n)}=k} \mathbf{z}_s^{(n)} + \sum_{q,n=1}^{Q,N} Q_{q,n}(k) \mathbf{z}_q^{(n)}}{\sum_{s,n=1}^{S,N} \mathbf{1}_{\mathbf{y}_s^{(n)}=k} + \sum_{q,n=1}^{Q,N} Q_{q,n}(k)}, \\ \boldsymbol{\sigma}_k^2 &:= \frac{\sum_{s,n=1}^{S,N} \mathbf{1}_{\mathbf{y}_s^{(n)}=k} (\mathbf{z}_s^{(n)} - \boldsymbol{\mu}_k)^2 + \sum_{q,n=1}^{Q,N} Q_{q,n}(k)(\mathbf{z}_q^{(n)} - \boldsymbol{\mu}_k)^2}{\sum_{s,n=1}^{S,N} \mathbf{1}_{\mathbf{y}_s^{(n)}=k} + \sum_{q,n=1}^{Q,N} Q_{q,n}(k)} \\ \psi &:= \{(\boldsymbol{\mu}_k, \boldsymbol{\sigma}_k^2, \frac{1}{K})\}_{k=1}^{K}, \end{aligned} \tag{10}$$

where $\mathbf{1}$ denotes an indicator function. We fix the mixing probability as $\frac{1}{K}$ since the labels in each task $\mathcal{T}$ are uniformly distributed. Moreover, we utilize diagonal covariance $\boldsymbol{\sigma}_k^2$ to obtain more accurate statistics for the inference. We initialize $\boldsymbol{\mu}_k$ and $\boldsymbol{\sigma}_k^2$ as the average value of support latent representations and the identity matrix $\boldsymbol{I}$, respectively. Similar to the meta-training stage, we obtain the MLE solution for the parameters of GMM, by performing E-step and M-step for a fixed number of iterations. Finally, we compute the conditional probability of $p(\mathbf{y}_q | \mathbf{x}_q, \mathcal{D})$ using the obtained parameters $\psi^*$ as follows:

$$p(\mathbf{y}_q | \mathbf{x}_q, \mathcal{D}) = \mathbb{E}_{q_\phi(\mathbf{z}_q | \mathbf{x}_q, \mathcal{D})} \left[ p_{\psi^*}(\mathbf{y}_q | \mathbf{z}_q) \right] \approx \frac{1}{N} \sum_{n=1}^{N} p_{\psi^*}(\mathbf{y}_q | \mathbf{z}_q^{(n)}), \quad \mathbf{z}_q^{(n)} \overset{i.i.d}{\sim} q_\phi(\mathbf{z}_q | \mathbf{x}_q, \mathcal{D}). \tag{11}$$

Here we compute $p_{\psi^*}(\mathbf{y}_q | \mathbf{z}_q^{(n)})$ with Bayes rule, and we reuse $N$ different Monte Carlo samples that is drawn for Eq 10, where the prediction of query $\hat{\mathbf{y}}_q = \arg\max_k p(\mathbf{y}_q = k | \mathbf{x}_q, \mathcal{D})$. We present the pseudo-code of the algorithm for training and inference of Meta-GMVAE in the Algorithm 1 and 2.

**Visual feature reconstruction** While our method is a generative model that can generate samples from output distribution, the ability to generate samples may not be necessary for discriminative downstream tasks (Chen et al., 2020). Moreover, we found that VAEs almost fail to learn in Mini-ImageNet dataset with the architecturally limited constraints of the meta-learning literature. Thus, we propose a high-level feature reconstruction objective instead for Mini-ImageNet dataset. We experimentally find that the recently proposed constrastive learning framework, namely SimCLR (Chen et al., 2020), is the most effective for our settings. Specifically, SimCLR learns high-level representation by performing a constrastive prediction task on pairs of augmented examples derived from a minibatch. We train SimCLR on the unsupervised dataset $\mathcal{D}_u = \{\mathbf{x}_u\}_{u=1}^U$, and use high-level features extracted by SimCLR as an input for our framework.

## 4 EXPERIMENT

In this section, we now validate the effectiveness of our Meta-GMVAE on several downstream few-shot classification tasks. The source codes are available at `https://github.com/db-Lee/Meta-GMVAE`.

### 4.1 EXPERIMENTAL SETUPS

**Baselines and ours** We now describe two supervised meta-learning approaches which we consider as "oracles", unsupervised meta-learning baselines, and the proposed Meta-GMVAE. **1) MAML (oracle)**: Model Agnostic Meta Learning by Finn et al. (2017). We compare against its performance reported in Hsu et al. (2019). **2) ProtoNets (oracle)**: Euclidean distance-based meta-learning approach by Snell et al. (2017). We also compare against it using its performance reported in Hsu et al. (2019). **3) CACTUs**: Clustering to Automatically Construct Tasks for Unsupervised meta-learning by Hsu et al. (2019). It automatically constructs tasks by clustering the unsupervised dataset in embedding space learned by ACAI (Berthelot et al., 2019), BiGAN (Donahue et al., 2017), and Deep-Cluster (Caron et al., 2018). Then they train either MAML or ProtoNets using the cluster indices as pseudo-labels. **4) UMTRA**: Unsupervised Meta-learning with Tasks constructed by Random sampling and Augmentation by Khodadadeh et al. (2019). For constructing a K-way 1-shot task, it randomly samples K-way datapoints from unsupervised dataset and augments each datapoint. Then MAML is trained on the constructed tasks. **5) Meta-GMVAE**: Our proposed Meta-level Gaussian Mixture VAE. It learns a latent representation by matching set-level amortized variational posterior and task-specific multimodal prior optimized by EM algorithm.

**Datasets** We validate all the models on two benchmark datasets for few-shot classification. **1) Omniglot**: This is a collection of $28 \times 28$ gray-scale hand-written characters that describe 1623 different alphabets, each of which contains 20 instances. Following the experimental setup of Hsu et al. (2019), we use 1200 classes for unsupervised meta-training, 100 classes for meta-validation and the remaining 323 classes for meta-test. We further augment each class by rotating the images 90, 180, and 270 degrees, such that the total number of classes is $1623 \times 4$, following the convention. **2) Mini-ImageNet**: This is a subset of ILSVRC-2012 (Deng et al., 2009) introduced by Ravi & Larochelle (2017), consisting of 100 classes that comes with 600 images of size $84 \times 84$ that describe different instances. We use 64 classes for unsupervised meta-training, 16 classes for meta-validation, and the remaining 20 classes for meta-test, following the standard protocol.

**Implementation details** We now introduce the specific implementation details of Meta-GMVAE on the two benchmark datasets. **1) Variational posterior network** $q_\phi(\mathbf{z}|\mathbf{x}, \mathcal{D}_i)$: we use the standard Conv4 architecture on Omniglot dataset for a fair comparison against relevant baselines. On top of the Conv4 architecture, we stack two TransformerEncoder layers and an affine transformation layer to predict the mean and log-variance of Gaussian distribution. For Mini-ImageNet dataset, we only utilize two TransformerEncoder layers and an affine transformation layer since the input used for Mini-ImageNet is already a high-level visual representation extracted from the Conv5 architecture trained with SimCLR. For both datasets, we set the dimensionality of the latent variable to 64. **2) Generative network** $p_\theta(\mathbf{x}|\mathbf{z})$: For Omniglot dataset, the architecture of generative network is symmetric to the Conv4 architecture of variational posterior network. The last layer outputs the parameter of output Bernoulli distribution. For Mini-ImageNet dataset, we use 3-layer MLP with ReLU activation to predict the mean of output Gaussian distribution. **3) Other details**: we utilize Adam optimizer (Kingma & Ba, 2015) with a constant learning rate of 0.001 and 0.0001 for Omniglot and MiniImageNet experiments, respectively. We set the number of iterations for EM algorithm as 10 for all the experiments. For the more details, please see the Appendix.

| | | Omniglot (way, shot) | | | | Mini-ImagNet (way, shot) | | | |
|---|---|---|---|---|---|---|---|---|---|
| Method | Clustering | (5,1) | (5,5) | (20,1) | (20,5) | (5,1) | (5,5) | (5,20) | (5,50) |
| *Training from Scratch* | N/A | 52.50 | 74.78 | 24.91 | 47.62 | 27.59 | 38.48 | 51.53 | 59.63 |
| CACTUs-MAML | BiGAN | 58.18 | 78.66 | 35.56 | 58.62 | 36.24 | 51.28 | 61.33 | 66.91 |
| CACTUs-ProtoNets | BiGAN | 54.74 | 71.69 | 33.40 | 50.62 | 36.62 | 50.16 | 59.56 | 63.27 |
| CACTUs-MAML | ACAI/DC | 68.84 | 87.78 | 48.09 | 73.36 | 39.90 | 53.97 | **63.84** | **69.64** |
| CACTUs-ProtoNets | ACAI/DC | 68.12 | 83.58 | 47.75 | 66.27 | 39.18 | 53.36 | 61.54 | 63.55 |
| UMTRA | N/A | 83.80 | 95.43 | 74.25 | **92.12** | 39.93 | 50.73 | 61.11 | 67.15 |
| **Meta-GMVAE (ours)** | N/A | **94.92** | **97.09** | **82.21** | 90.61 | **42.82** | **55.73** | 63.14 | 68.26 |
| *MAML (oracle)* | N/A | 94.46 | 98.83 | 84.60 | 96.29 | 46.81 | 62.13 | 71.03 | 75.54 |
| *ProtoNets (oracle)* | N/A | 98.35 | 99.58 | 95.31 | 98.81 | 46.56 | 62.29 | 70.05 | 72.04 |

Table 1: The few-shot classification results (way, shot) on the Omniglot and Mini-ImageNet datasets. DC denotes DeepCluster. We report the average of accuracies evaluated over 1000 episodes. All the values are based on the reported performance in Hsu et al. (2019) and Khodadadeh et al. (2019), except for ours.

(a) Meta-train (Real)    (b) Meta-train (Generated)    (c) Meta-test (Real)    (d) Meta-test (Generated)

Figure 3: The samples obtained and generated for each mode at unsupervised meta-training and supervised meta-test step of Meta-GMVAE. Samples in each row are in the same modality obtained by EM.

## 4.2 EXPERIMENTAL RESULTS

**Few-shot classification** Table 1 shows the few-shot classification results obtained by supervised meta-learning baselines (oracle), the two unsupervised meta-learning baselines, and our Meta-GMVAE. For the Omniglot dataset, the Meta-GMVAE outperforms all the baselines that only utilize unsupervised-learning for constructing meta-training tasks, except for the UMTRA on the 20-shot 5-shot classification. Meta-GMVAE also outperforms baselines on Mini-ImageNet 1-shot, and 5-shot settings which are the most widely used settings, while it matches the performance of baselines in 20-shot, and 50-shot settings. This shows that meta-learning the posterior network can capture the multi-modal distribution of any given tasks with Meta-GMVAE, is indeed more effective over unsupervised meta-learning baselines which simply trains supervised meta-learning models with pseudo-labels obtained from unlabeled data. Moreover, our Meta-GMVAE obtains better performance than supervised MAML on Omniglot 5-way 1-shot classification, while utilizing as small as $0.1\%$ of the labeled data. This matches the observation in Chen et al. (2020) that well-calibrated unsupervised learning approaches with a modest amount of labels can obtain a performance comparable to or even better than supervised approaches.

**Visualization** To better understand how our Meta-GMVAE learns and realizes class-concepts in few-shot classification tasks, we visualize the actual samples in an episode classified by Meta-GMVAE and ones generated by generative network $p_\theta(\mathbf{x}|\mathbf{z})$, during unsupervised meta-training and supervised meta-testing. We visualize the actual samples and generated ones that have a same modality in a same row. In Figure 3-**a, b**, we can observe that our Meta-GMVAE captures the similar visual structure in each modality during meta-training, but the modalities are not the class-concepts. However, as shown Figure 3-**c, d**, our Meta-GMVAE easily realizes each modality as each class-concept at meta-test time.

**Ablation study** Furthermore, we compare the performance of our model variants by eliminating each of the most important components for our model. We describe the each variant as follows: **1) LR (SimCLR)**: This performs the logistic regression using support set on top of features pretrained by SimCLR. **2) Vanilla VAE**: We train Vanilla VAE on $\mathcal{D}_u$ and predict labels using semi-supervised EM with fixed identity covariance $I$. **3) Vanilla VAE (SimCLR)**: This is same as **2) Vanilla VAE** except that it is trained on features pretrained by SimCLR. **4) Ep**: Meta-GMVAE with an episodic training with task specific parameter $\psi_i^*$ obtained by EM. **5) Set**: Meta-GMM whether having set-

| Method | Ep | Set | $\sigma^2$ | O | M | | Training (way, shot) | | | |
|--------|----|----|-----------|---|---|---|---|---|---|---|
| | | | | | | **Test way** | **(5, 1)** | **(5, 5)** | **(20, 1)** | **(20, 5)** |
| *Training From Scratch* | | | | 24.91 | 27.59 | 2-way | 98.26 | 98.00 | 98.36 | 98.23 |
| LR (SimCLR) | | | | N/A | 40.11 | 5-way | 94.92 | 94.57 | 93.93 | 94.01 |
| Vanilla VAE | | | | 69.68 | N/A | 10-way | 89.87 | 89.99 | 89.10 | 89.30 |
| Vanilla VAE (SimCLR) | | | | N/A | 38.40 | 15-way | 85.11 | 85.12 | 85.36 | 85.33 |
| | ✓ | | | 78.64 | 40.51 | 20-way | 81.38 | 81.11 | 82.21 | 81.98 |
| Meta-GMVAE | ✓ | ✓ | | 81.65 | 41.13 | 30-way | 77.80 | 77.42 | 78.40 | 77.24 |
| | ✓ | | ✓ | 80.94 | 40.92 | 40-way | 73.76 | 73.15 | 74.03 | 73.56 |
| | ✓ | ✓ | ✓ | **82.21** | **42.82** | 50-way | 70.92 | 70.85 | 69.86 | 70.02 |
| *MAML (oracle)* | ✓ | | | 84.60 | 46.81 | | | | | |
| *ProtoNets (oracle)* | ✓ | | | 95.31 | 46.56 | | | | | |

Table 2: **Left**: The results of the ablation study on Meta-GMVAE (**O**: 20-way 1-shot classification on Omniglot, **M**: 5-way 1-shot classification on Mini-ImageNet). **Right**: The results of cross-way 1-shot experiments on Omniglot. The values in the parenthesis indicate that a model is trained based on the (way, shot) setting.

level variational posterior (i.e. $q_\phi(\mathbf{z}|\mathbf{x}, \mathcal{D}_i)$) or not (i.e. $q_\phi(\mathbf{z}|\mathbf{x})$). **6)** $\sigma^2$: Meta-GMM performs semi-supervised EM algorithm whether using diagonal covariance matrix or fixing it with identity matrix $\boldsymbol{I}$. Table 2-**Left** shows that all the components we consider are critical for the performance on the few-shot classification tasks as expected. The best performance gain comes from **Ep**, which supports our proposal on meta-learning the set-level variational posterior by matching it with the multi-modal prior, where the task-specific parameter is obtained with EM.

**Cross-way classification** We then experiment our Meta-GMVAE by varying the number of way (between 2 and 50) and fixing the number of shot as 1. In particular, we set the number of component $k$ as the **Test way** for the meta-test and perform semi-supervised EM algorithm in Eq 10. Table 2-**Right** shows that the difference in the number of way used for training and test does not significantly affect the performance, which demonstrates the robustness of Meta-GMVAE on varying number of way. We

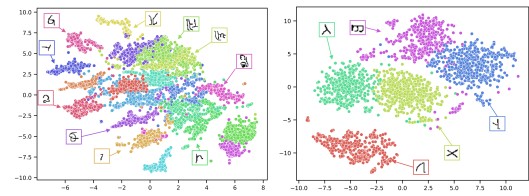

(a) 20-way Meta-training  (b) 5-way Meta-test

Figure 4: The visualization of the latent space for the cross-shot generalization experiment.

also visualize the latent space for the cross-shot experiment using t-SNE (Rauber et al., 2016), in Figure 4, which shows that Meta-GMVAE trained with 20-way can cluster 5-way meta-test task.

## 5 CONCLUSION

We proposed a novel unsupervised meta-learning model, namely Meta-GMVAE, which can generate a task-dependent posterior for a given unseen task with multi-modal Gaussian Mixture priors. Given a random episode that consists of samples from diverse classes, we optimize the task-specific parameter of the mixture of Gaussian prior with Expectation-Maximization algorithm, such that each mode can capture intrinsic groupings in the given data. We meta-train the variational posterior network over such data-driven prior obtained over large number of episodes. Then, at the meta-test step, we realize each modality with a label by deploying semi-supervised EM algorithm with both the support and the query set. We validate our method on two few-shot image classification benchmark datasets, and show that Meta-GMVAE largely outperforms the relevant unsupervised meta-learning baselines, even achieving better performance than supervised MAML on Omniglot 5-way 1-shot experiments.

**Acknowledgements** This work was supported by Institute of Information & communications Technology Planning & Evaluation (IITP) grant funded by the Korea government (MSIT) (No.2019-0-00075, Artificial Intelligence Graduate School Program (KAIST)), Samsung Research Funding Center of Samsung Electronics (No. SRFC-IT1502-51), Samsung Electronics (IO201214-08145-01), and the Engineering Research Center Program through the National Research Foundation of Korea (NRF) funded by the Korean Government MSIT (NRF-2018R1A5A1059921). We sincerely thank the anonymous reviewers for their constructive comments which helped us significantly improve our paper during the rebuttal period. We also appreciate D. Khuê Lê-Huu for the valuable discussion on Rao et al. (2019).

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

# A  OMNIGLOT EXPERIMENTS

## A.1  TRAINING PROCEDURE

Omniglot is a collection of $28 \times 28$ gray-scale hand-written characters that describe 1623 different alphabets, each of which contains 20 instances. Following the experimental setup of Hsu et al. (2019), we use 1200 classes for unsupervised meta-training, 100 classes for meta-validation and the remaining 323 classes for meta-test. We further augment each class by rotating the images 90, 180, and 270 degrees, such that the total number of classes is $1623 \times 4$, following the convention. We evaluate the trained model using 1000 randomly selected tasks from test set. During evaluation, $K \times S$ data instances are used as support inputs and $K \times 15$ data instances are used as query inputs. We use the Adam (Kingma & Ba, 2015) optimizer with a constant learning rate of 0.001 to train all models. All models are trained for 60,000 iterations. For the 5-way experiments (i.e. $K = 5$), we set the mini-batch size, the number of datapoints, and Monte Carlo sample size as 4, 200, and 32, respectively (i.e. $B = 4, M = 200$, and $N = 32$). For the 20-way experiments (i.e. $K = 20$), we set them as 4, 300, and 32 (i.e. $B = 4, M = 300$, and $N = 32$). We set the number of EM iterations as 10.

## A.2  NETWORK ARCHITECTURE

We summarize the network architecture in the following Table 3, and 4. We assume that the output follows Bernoulli distribution, therefore, the output of generative network $p_\theta(\mathbf{x}|\mathbf{z})$ is the mean parameter.

**Set-level variational posterior network $q_\phi(\mathbf{z}|\mathbf{x}, \mathcal{D}_i)$**

| Output Size | Layers |
|---|---|
| $1 \times 28 \times 28$ | Input Images |
| $64 \times 14 \times 14$ | conv2d($3 \times 3$, stride 1, padding 1), BatchNorm2D, ReLU, Maxpool($2 \times 2$, stride 2) |
| $64 \times 7 \times 7$ | conv2d($3 \times 3$, stride 1, padding 1), BatchNorm2D, ReLU, Maxpool($2 \times 2$, stride 2) |
| $64 \times 4 \times 4$ | conv2d($3 \times 3$, stride 1, padding 1), BatchNorm2D, ReLU, Maxpool($2 \times 2$, stride 2) |
| $64 \times 2 \times 2$ | conv2d($3 \times 3$, stride 1, padding 1), BatchNorm2D, ReLU, Maxpool($2 \times 2$, stride 2) |
| $256$ | Flatten |
| $256$ | TransformerEncoder($d_{\text{model}} = 256, d_{\text{ff}} = 256, h = 4$, ELU, LayerNorm = False) |
| $256$ | TransformerEncoder($d_{\text{model}} = 256, d_{\text{ff}} = 256, h = 4$, ELU, LayerNorm = False) |
| $64 \times 2$ | Linear($256, 64 \times 2$) |

Table 3: Set-level variational posterior network used for Omniglot dataset. We refer the hyperparameter notation of TransformerEncoder to Vaswani et al. (2017).

**Generative network $p_\theta(\mathbf{x}|\mathbf{z})$**

| Output Size | Layers |
|---|---|
| $64$ | Latent code |
| $256$ | Linear($64,256$), ELU |
| $256$ | Linear($256, 256$), ELU |
| $256$ | Linear($256, 256$), ELU |
| $64 \times 2 \times 2$ | Unflatten |
| $64 \times 4 \times 4$ | deconv2d($4 \times 4$, stride 2, padding 1), BatchNorm2D, ReLU |
| $64 \times 7 \times 7$ | deconv2d($3 \times 3$, stride 2, padding 1), BatchNorm2D, ReLU |
| $64 \times 14 \times 14$ | deconv2d($4 \times 4$, stride 2, padding 1), BatchNorm2D, ReLU |
| $1 \times 28 \times 28$ | deconv2d($4 \times 4$, stride 2, padding 1), Sigmoid |

Table 4: Generative Network for $p_\theta(\mathbf{x}|\mathbf{z})$ for Omniglot dataset.

## A.3  95% CONFIDENCE INTERVAL

We provide the standard errors of our model's performance at 95% confidence interval over 1000 episodes on the Omniglot dataset in Table 5.

| Omniglot | (5,1) | (5,5) | (20,1) | (20,5) |
|---|---|---|---|---|
| **Meta-GMVAE** | $94.92 \pm 0.42$ | $97.09 \pm 0.20$ | $82.21 \pm 0.44$ | $90.61 \pm 0.19$ |

Table 5: The few-shot classification results (way, shot) with 95% confidence interval on the Omniglot.

## B  MINI-IMAGENET EXPERIMENTS

### B.1  TRAINING PROCEDURE

Mini-ImageNet is a subset of ILSVRC-2012 (Deng et al., 2009) introduced by Ravi & Larochelle (2017), consisting of 100 classes that comes with 600 images of size $84 \times 84$ that describe different instances. We first train Conv5 feature extractor using SimCLR objective with temperature term $\tau = 0.5$, on Mini-ImageNet unsupervised meta-training dataset. We train the feature extractor using Adam optimizer with learning rate of 0.0001 for 400 epochs. We use 64 classes for unsupervised meta-training, 16 classes for meta-validation, and the remaining 20 classes for meta-test, following the standard protocol. We evaluate the trained model using 1000 randomly selected tasks from test set. During evaluation, $5 \times S$ data instances are used as support inputs and $5 \times 15$ data instances are used as query inputs. For all the experiments, we use the Adam (Kingma & Ba, 2015) optimizer with a constant learning rate of 0.0001, and set the mini-batch size, the number of datapoints, and Monte Carlo sample size as 16, 5, and 256, respectively (i.e. $B = 16, M = 5$, and $N = 256$) for the 1, 5, and 20-shot experiments. For the 50-shot experiment, we set them 4, 200, and 256, respectively (i.e. $B =, M = 200$, and $N = 256$). We train the models for 5K, 10K, 15k, 25K, and 30K for 1, 5, 20, and 50-shot experiments, respectively. We set the number of EM iterations as 10.

### B.2  NETWORK ARCHITECTURE

We summarize the network architecture in the following Table 6, 7, and 8. We assume that the output follows Gaussian distribution, therefore, the output of generative network $p_\theta(\mathbf{x}|\mathbf{z})$ is the mean parameter. Moreover, the variance of output Gaussian distribution is obtained as suggested in Rybkin et al. (2020).

**Feature Extractor for SimCLR**

| Output Size | Layers |
|---|---|
| $3 \times 84 \times 84$ | Input Images |
| $64 \times 42 \times 42$ | conv2d($3 \times 3$, stride 1, padding 1), BatchNorm2D, ReLU, Maxpool($2 \times 2$, stride 2) |
| $64 \times 21 \times 21$ | conv2d($3 \times 3$, stride 1, padding 1), BatchNorm2D, ReLU, Maxpool($2 \times 2$, stride 2) |
| $64 \times 10 \times 10$ | conv2d($3 \times 3$, stride 1, padding 1), BatchNorm2D, ReLU, Maxpool($2 \times 2$, stride 2) |
| $64 \times 5 \times 5$ | conv2d($3 \times 3$, stride 1, padding 1), BatchNorm2D, ReLU, Maxpool($2 \times 2$, stride 2) |
| $64 \times 2 \times 2$ | conv2d($3 \times 3$, stride 1, padding 1), BatchNorm2D, ReLU, Maxpool($2 \times 2$, stride 2) |
| 256 | Flatten |

Table 6: Feature Extractor trained on Mini-ImageNet dataset using SimCLR objective.

**Set-level variational posterior network $q_\phi(\mathbf{z}|\mathbf{x}, \mathcal{D}_i)$**

| Output Size | Layers |
|---|---|
| 256 | Input Features |
| 256 | TransformerEncoder($d_{\text{model}} = 256, d_{\text{ff}} = 256, h = 4$, ReLU, LayerNorm = False) |
| 256 | TransformerEncoder($d_{\text{model}} = 256, d_{\text{ff}} = 256, h = 4$, ReLU, LayerNorm = False) |
| $64 \times 2$ | Linear(256, $64 \times 2$) |

Table 7: Set-level variational posterior network used for Mini-ImageNet dataset. We refer the hyperparameter notation of TransformerEncoder to Vaswani et al. (2017).

| Generative network $p_\theta(\mathbf{x}|\mathbf{z})$ | |
|---|---|
| **Output Size** | **Layers** |
| 64 | Latent code |
| 512 | Linear(64, 512), ReLU |
| 512 | Linear(512, 512), ReLU |
| 256 | Linear(512, 256), ReLU |

Table 8: Generative Network for $p_\theta(\mathbf{x}|\mathbf{z})$ for Mini-ImageNet dataset.

## B.3 95% CONFIDENCE INTERVAL

We provide the standard errors at 95% confidence interval over 1000 episodes on the Mini-ImageNet dataset in Table 9.

| Mini-ImageNet | (5,1) | (5,5) | (20,1) | (20,5) |
|---|---|---|---|---|
| **Meta-GMVAE** | $42.82 \pm 0.56$ | $55.73 \pm 0.48$ | $63.14 \pm 0.47$ | $68.26 \pm 0.42$ |

Table 9: The few-shot classification results (way, shot) with 95% confidence interval on the Mini-ImageNet.

## B.4 ADDITIONAL COMPARISON USING SIMCLR

To further understand where the improvement of Meta-GMVAE on the Mini-ImageNet dataset, we ran experiment on baselines with SimCLR pretrained features. For CACTUs, we cluster in the embedding space pretrained by SimCLR. For UMTRA, we follow the exact same procedure to generate training episode, which is proposed by the authors. Moreover, we fix the pretrained SimCLR features as the setting of Meta-GMVAE for the both of baselines. Table 10 shows that Meta-GMVAE outperforms the baselines with SimCLR, which supports the effectiveness of Meta-GMVAE combined with SimCLR pretrained features.

| Mini-ImageNet | (5,1) | (5,5) | (20,1) | (20,5) |
|---|---|---|---|---|
| **CACTUs-MAML (SimCLR)** | 40.39 | 52.35 | 61.09 | 64.89 |
| **UMTRA (SimCLR)** | 40.85 | 51.47 | 61.03 | 67.30 |
| **Meta-GMVAE** | 42.82 | 55.73 | 63.14 | 68.26 |

Table 10: The comparison on the few-shot classification results (way, shot) using SimCLR.

