# OpenReview forum: "Meta-GMVAE: Mixture of Gaussian VAE for Unsupervised Meta-Learning"
_ICLR.cc/2021/Conference — ICLR 2021 Spotlight_

### Official Review · AnonReviewer1 · 2020-10-21
**Simple and effective extension of Mixture of Gaussians to a VAE model to perform meta-learning.  Results are good, but novelty seems lacking.**

**Rating:** 7
**Confidence:** 4

**Review:**

The paper goal is to learn unsupervised feature representations that can be transferred between few-shot classification tasks.  The paper models the class-concepts with a Mixture of Gaussians prior, and uses Variational Autoencoders to model the latent representations between the tasks and the samples.

The presentation is clear and straightforward.  The idea is to use a GMM and use an Expectation-Maximization (EM) approach to learn the mixture.  To tackle the intractability of the variational posterior $q_\phi (z_j | x_j, \mathcal{D}_i)$, the paper proposes to use a Monte Carlo approximation.  For the meta-test, the model is tuned using EM in a semi-supervised fashion.

The experiments show the superiority of the Meta-GMVAE and the compared methods on the Omniglot and Mini-ImageNet datasets.

Nevertheless, I find the idea simple, yet compelling.  The idea of adding GMM to enhance the modeling capabilities is a well known fact, and that has been explored before.  For instance, some recent publications applying the GMM idea (not that the final application and overall implementation may differ from meta-learning---see my comment below):
- Dilokthanakul et al., Deep Unsupervised Clustering with Gaussian Mixture Variational Autoencoders, https://arxiv.org/abs/1611.02648
- Zhao et al., Truncated Gaussian-Mixture Variational AutoEncoder, https://arxiv.org/abs/1902.03717
- Guo et al., Variational Autoencoder With Optimizing Gaussian Mixture Model Priors, 10.1109/ACCESS.2020.2977671.
- Yang et al., Deep Clustering by Gaussian Mixture Variational Autoencoders with Graph Embedding, 10.1109/ICCV.2019.00654

It seems from the presentation that the main difference is the application to the meta-learning approach.  The authors should explain better what the contribution is and how it contrast to the existing literature of mixture models applied in variational modeling.

Pros:
- Simple and effective idea.
- Use of well known methods with simple approximators.
- Good results on the presented experiments.

Cons:
- The contribution is not clear.  I'm on the fence of whether the usage of the GMM to a new task is enough to guarantee a publication.

Overall rating: ~~I'm giving a 5 due to the lack of clarity in the contribution and added novelty.  However, the presentation is good, and the explanations are clear.~~
I'm updating my rating to accept the paper due to the comments and updates on the paper.  The proposed flexible usage of the GMM is novel from the existing literature.  The changes in the paper improved its clarity, and the contribution is better presented in contrast to existing work.

---

> ### Author Response · Authors · 2020-11-13
> **Regarding the lack of clarity in the contribution.**
>
> We appreciate your constructive comments. We respond your main concerns below:
>
> It seems from the presentation that the main difference is the application to the meta-learning approach. The contribution is not clear.
>
> - The related works you mentioned mainly aim to use GMM for modeling the variational posterior or prior to enhance model capacity, for a single task. On the other hand, the main contribution of our work is the modeling a meta-network that can adaptively generate a GMM prior for any given tasks (datasets). We strongly believe that this is a highly novel direction in the perspective of both the **unsupervised meta-learning** we target, and the **modeling of the GMM posterior/prior**.
>
> ---
>
> - First of all, the previous works you mentioned that utilize GMMs consider **single task learning**, therefore, the learned GMM parameterized with a neural network is **fixed after training**. However, in our unsupervised meta-learning setting, a model should be able to adapt and generalize to a novel task, which is not possible with the existing approaches for learning GMMs. To achieve such a flexibility, we propose to model the GMM **prior in a set-dependent manner**, by adapting it to a new dataset with the **variational posterior conditioned on both the dataset and each data point**. Specifically, the parameter of GMM prior flexibly adapts to a given dataset, by optimizing the local variational lower bound with respect to the parameter using EM algorithm. In this respect, our method is novel over existing approaches in the GMM VAE literature.
>
> ---
>
> - Secondly, in the context of unsupervised meta-learning, existing approaches on the topic relied on the strategy of constructing supervised meta-training tasks with pseudo-labels, either by clustering the data (CACTUs), or assign each data point from a randomly sampled batch to a pseudo-class (UMTRA). Thus, they are **essentially the same as the supervised meta-learning** approaches except for the **pseudo-labeling strategies**. On the contrary, we propose a **principled unsupervised meta-learning** model based on a VAE framework with meta-learned GMM priors, which we believe is based on a completely new paradigm.
>
> ---
>
> We hope that the above discussions clear up your concern regarding the novelty, and will be happy to further respond to any more questions or comments.

---

> > ### Comment · AnonReviewer1 · 2020-11-17
> > **The authors clarified my doubts**
> >
> > I thank the authors for the thorough response.  Indeed, under this new light the method explores a new and interesting modelling avenue for unsupervised meta-learning.
> >
> > Moreover, I would suggest the authors to add a sentence or two to the contributions at the end of the Introduction to, also, clarify the flexibility of the proposed posterior in comparison to existing approaches.
> >
> > Due to the changes in the manuscript that improve its clarity and the main contribution, I will upgrade my rating.

---

> > > ### Author Response · Authors · 2020-11-17
> > > **Regarding the suggestion to add contributions at the end of the Introduction**
> > >
> > > We sincerely appreciate that you find our work as "exploring a new and interesting modeling avenue for unsupervised meta-learning". As you suggested, we have revised the first and second bullet points of the contributions at the end of the Introduction section of the **revision**as follows:
> > >
> > > ---
> > >
> > > **1. First bullet point**
> > >
> > > - **Original**: We propose a novel unsupervised meta-learning model which meta-learns the posterior network of a VAE, which is a principled unsupervised meta-learning method unlike existing unsupervised meta-learning methods that combines heuristic pseudo-labeling with supervised meta-learning.
> > >
> > > - **Modified**: We propose a novel unsupervised meta-learning model, namely Meta-GMVAE, which meta-learns the set-conditioned prior and posterior network for a VAE. Our Meta-GMVAE is a principled unsupervised meta-learning method, unlike existing methods on unsupervised meta-learning that combines heuristic pseudo-labeling with supervised meta-learning.
> > >
> > > ---
> > >
> > > **2. Second bullet point**
> > >
> > > - **Original**: We propose to capture multi-modal structure in the given dataset as Gaussian mixtures, using EM algorithm at meta-training and semi-supervised EM at meta-test time.
> > >
> > > - **Modified**: We propose to learn the multi-modal structure of a given dataset with the Gaussian mixture prior, such that it can adapt to a novel dataset via the EM algorithm. This flexible adaptation to a new task, is not possible with existing methods that propose VAEs with Gaussian mixture priors for single task learning.
> > >
> > > ---

---

### Official Review · AnonReviewer4 · 2020-10-26
**Good paper, I hope for follow-ups without VAEs**

**Rating:** 8
**Confidence:** 3

**Review:**

The problem which the authors attempt to solve is unsupervised meta-learning (UML), ie. learning in an unsupervised way such a model of a dataset, as to be able to perform meta-learning (here: few-shot classification) later. I see their contribution as two-fold:
1. Proposing a framework for solving UML consisting of sampling subsets $D_i$ of a full dataset $D_u$, training a generative model based on both datapoints ($x_j$) themselves and the particular subset $D_i$ and using it in a semi-supervised fashion.
2. Implementing a model in this framework based on a VAE. Here, the latent variable $z$ doesn't just compress information about a datapoint (as in a classical VAE), but is also able to encode (in an abstract way) the position of this datapoint in the subset $D_i$ (ie. "task-specific label"). To be able to capture this (arguably richer than in classical VAEs) distribution, authors use a GMM to model the variational distribution.

Because MLE of GMM is intractable, authors have a two stage optimization process:
a) Finding a task-specific (ie. encoding info about $D_i$ "classes") parameter $\phi^*$ via EM and b) Optimizing the ELBO given $\phi^*$ as usual. During meta-testing, the $\phi^*$ parameter is estimated in a similar way using the test-time samples $x_i$ (trying to embed the new task into the learned manifold) and then latent variable $z$ is sampled conditionally based on $x_i$ and the expected value of the constructed distribution $p_{\phi^*}(y|z)$ estimated via Monte Carlo.

1. While I am neither a VAE expert nor enthusiast, I consider the proposed model principled: while the two-stage optimization mechanism is not ideal (as may make it harder to optimize compared to end-to-end differentiable models), learning a single distribution describing both elements we care about: images and their placement within a dataset seem to match the problem better than previous pseudo-labels-based methods.
2. I particularly like introduction of the general framework (1.) (which is not emphasized in the paper). I believe that it should be possible (not necessarily straight-forwardly) to extend the proposed model to other generative models. To make it clear, I wouldn't expect this extension from the paper under review (what I'm proposing is basically yet another paper), but the opening of this direction of research is a big plus.
3. Paper is easy to understand.
4. The presented results, while competitive compared to the previous UML SOTA, are only presented on somewhat toyish problems (Omniglot, mini-imagenet). While it is understandable that it'll be hard to train a Meta-GMVAE on more complex datasets (as it's only harder than classic VAEs, which are already struggling with higher-dimensional tasks), presenting the results only on small datasets (even if this is the current SOTA and other methods do it) somewhat undermines the overall motivation to UML: to be able to use vast amounts of unstructured data while building ML models.
5. I am not able to comment on the novelty of the work: I am barely aware of the contemporary VAE/UML literature. I will be willing to modify my score based on other reviewers' opinions in that regard.

Question/proposal:
In Sec. 3.2. authors write "assuming that the modalities in prior distribution represent class-concepts of any datasets". Why would this be the case? This seems intuitive; I feel like there could be a nice theoretical argument why it would be the case.

I find the model principled and new. It solves an important problem in a natural way, improving over SOTA and opening the potential for follow-up research. I weakly question the use of VAEs, which feels like it is limiting the method (making hi-dim UML impossible), but am aware that is more of a complaint against a well-established research domain than the contribution of this paper itself.

Typos:
1. Abstract:
... from unlabeled data which can capture ...
... shares the spirit of unsupervised learning in that they both seek ...
2. Sec. 1:
... effectiveness of our framework, we run experiments on ...
3. Sec. 2:
... One of the main limitations of ...
4. Sec. 3.2.:
... inferring isotropic Gaussian distribution, to encode ...

---

> ### Author Response · Authors · 2020-11-14
> **Response**
>
> We sincerely appreciate your constructive comments. We respond to your main concerns below:
>
> **1.** the two-stage optimization mechanism is not ideal
> - We agree that the two-stage optimization mechanism could introduce some instability. However, please note that we perform EM in the latent embedding space, which is a relatively low-dimensional space EM algorithm is known to work.
>
> ---
>
> **2.** I believe that it should be possible (not necessarily straight-forwardly) to extend the proposed model to other generative models.
> - This is an insightful comment. The derivation of Eq. (1) is already similar to other generative models such as **Neural Statistician [1]**, which models the log likelihood of a dataset with VAE framework. However, Neural Statistician assumes that each dataset consists of data points in the same class, which is only possible with supervised learning while we tackle unsupervised learning scenarios. We have included discussions of the Neural Statistician in the related work section of the revision.
>
> ---
>
> **3.** The presented results, while competitive compared to the previous UML SOTA, are only presented on somewhat toyish problems (Omniglot, mini-imagenet).
> - Thank you for the suggestion and we will consider more complex datasets as future work. Please also note that the two datasets are standard benchmark datasets for meta-learning, and we had to use them to compare against existing works since all baselines use the two datasets for performance evaluation.
>
> ---
>
> **4.** Concern on novelty of Meta-GMVAE.
> - We strongly believe that Meta-GMVAE is a highly novel work both in the perspective of the **unsupervised meta-learning** we target, and the modeling of **the GMM posterior/prior** for VAE.
> - First of all, existing unsupervised meta-learning methods focused on constructing pseudo-labeled dataset and then simply apply supervised meta-learning models on the pseudo-labeled datasets. Thus, they are essentially the same as the supervised meta-learning approaches. To our knowledge, our work is the first **principled unsupervised meta-learning** model which meta-learns the multi-modal prior in a task (dataset)-dependent manner.
> - Secondly, our work is largely different from previous work on the modeling of the GMM posterior/prior since we model the GMM prior in a **dataset-dependent manner**. This makes Meta-GMVAE can adapt to unseen tasks which is the most important property of meta-learning models.
>
> ---
>
> **5.** "assuming that the modalities in prior distribution represent class-concepts of any datasets". Why would this be the case?
> - As far as we know, previous work in the literature of the modeling GMM for VAE [2,3,4,5] (suggested by R1) usually aims at unsupervised clustering which is evaluated by a classification accuracy. This shares the same idea that each cluster might represent a class-concept. In our case, the each cluster **explicitly** represents a class-concept in the meta-test since we perform semi-supervised EM algorithm in Eq (8) using the labels of support set.
>
> ---
>
> **6.** I weakly question the use of VAEs, which feels like it is limiting the method (making hi-dim UML impossible).
> - We agree with your concern on the limitations of VAEs. However, please note that we can use VAEs to generate latent features rather than images as we did with the miniImageNet dataset. Moreover, recent works such as [6] have successfully implemented VAEs for generating super high-resolution images. We believe that combining such recent advances in VAEs with ours can be another promising research direction to enable high-dimensional UML.
>
> ---
>
> Typos
>
> - Thank you for pointing them out. We have revised them in the revision.
>
> ---
>
> References:
>
> [1] [Harrison Edwards, and Amos Storkey. "Towards a Neural Statistician." In ICLR, 2017.](https://arxiv.org/pdf/1606.02185.pdf)
>
> [2] [Dilokthanakul et al. "Deep Unsupervised Clustering with Gaussian Mixture Variational Autoencoders." In arXiv, 2016](https://arxiv.org/abs/1611.02648)
>
> [3] [Zhao et al. "Truncated Gaussian-Mixture Variational AutoEncoder." In IPMI 2019.](https://arxiv.org/abs/1902.03717)
>
> [4] Guo et al. "Variational Autoencoder With Optimizing Gaussian Mixture Model Priors." in IEEE 2020
>
> [5] [Yang et al. "Deep Clustering by Gaussian Mixture Variational Autoencoders with Graph Embedding." In ICCV 2019.](https://openaccess.thecvf.com/content_ICCV_2019/papers/Yang_Deep_Clustering_by_Gaussian_Mixture_Variational_Autoencoders_With_Graph_Embedding_ICCV_2019_paper.pdf)
>
> [6] [Arash Vahdat, and Jan Kautz. "NVAE: A Deep Hierarchical Variational Autoencoder." In arXiv 2020.](https://arxiv.org/pdf/2007.03898.pdf)

---

### Official Review · AnonReviewer3 · 2020-11-02
**The semi-supervised meta-learning setting considered in the paper is interesting, and the use of an embedded mixture model to construct pseudo labels in this setting is a nice development. However, the submission lacks clarity on and motivation for the components of the algorithm, and there are some concerns with the empirical evaluation and the absence of references to some very relevant work.**

**Rating:** 7
**Confidence:** 5

**Review:**

The submission proposes an algorithm for the semi-supervised meta-learning (unsupervised meta-training + supervised meta-testing) setting of [1], which adapts the few-shot learning + evaluation setting of [2, 3] by omitting classification labels at meta-training time. The algorithm makes use of a variational auto-encoder (VAE) formulation defined over a hierarchical model that describes the decomposition of a dataset into tasks of datapoint-target pairs (i.e., the meta-learning setup). The prior distribution of the hierarchical VAE is taken to be a mixture of Gaussians to facilitate the construction of pseudo-labels at meta-training time. The algorithm is evaluated on the Omniglot and miniImageNet few-shot classification tasks (with labels unused at meta-training time).

##### Strengths:

The semi-supervised meta-learning (unsupervised meta-training + supervised meta-testing) setting is interesting and worthy of study as an analogue of unsupervised learning. The datasets used in the empirical evaluation are appropriate, although they do not represent the most complex image datasets used in few-shot classification evaluations (cf. meta-dataset [8]).

The use of a Gaussian mixture model (GMM) for the prior distribution of a variational auto-encoder (VAE), which allows an analytic solution for a subset of the variational parameters, is conceptually interesting, although its use would not be restricted to the meta-learning setting. Using it to construct pseudo-labels (as well as incorporate labels when available) for the semi-supervised meta-learning setting is a nice development, although not a significant advance from the use of $k$-means in CACTUs.

Meta-GMVAE attaints higher performance than semi-supervised few-shot classification setting comparison methods (CACTUs & UMTRA) on the Omniglot and miniImageNet benchmarks; moreover, it approaches the level of a supervised few-shot classification method (MAML) on the Omniglot benchmark (although this supervised comparator does not represent state-of-the-art performance on this benchmark).

##### Weaknesses:

1) **Clarity**: The algorithmic components of the submission were very difficult to get straight. In the development of the algorithm in Section 3.2, the submission does not adequately discuss why and how particular subcomponents are employed, and various points about the different algorithmic components are made in the text without sufficient explanation or integration; some examples are:

    - The VAE formulation is introduced without precedent just above equation (1). It is also a bit of a red herring because it is not subsequently used, as is, in the algorithm.

    - At "The difference of our model from original VAE is that we utilize a set-level variational posterior $q_\phi(\mathbf{z}_j |\mathbf{x}_j , D_i)$, for inferring isotropic Gaussian distribution, to encode characteristics of a given dataset $D_i$. Specifically, we utilize self-attention mechanism (Vaswani et al., 2017) on top of a convolutional neural network." This is the first time an "isotropic Gaussian distribution" is mentioned in the method, self-attention is not explained further, and there is no explanation of how the convolutional neural network (CNN) fits into the whole framework. For example, it is not clear from this section whether (and if so, how) a CNN is used in addition to the SimCLR feature representation.

    - "...we set the prior distribution as a mixture of Gaussians (GMM), where $y$ is a discrete random variable indicating the component of a latent variable $\mathbf{z}$". $y$ and $\mathbf{z}$ are not yet defined except by reference to the VAE formulation in (1), but that was insufficiently explained as a part of the algorithm.

    More specific details for reproducibility are not described in the text (e.g., how the GMM parameters are initialized for EM; what some of the variables ($\mathbf{z}$, $\mathbf{x}$, $\phi$, $\theta$) refer to in the implementation). On top of this, results would be extremely difficult to reproduce: While component architectures and experimental setups are detailed in the appendix, how everything fits together is not adequately described.

    More broadly, the submission would benefit significantly from an algorithm box to convey how all the components interact and which components act episodically (at the task level) vs. at the level of the entire dataset.

2) **Quality**: The experimental evaluation does not provide a measure of variance (e.g., 95% confidence interval) in Table 1, which should be provided to ascertain the significance of the reported improvement.

  The algorithm uses the SimCLR representation learning objective to pre-train the feature extractor, while the comparison semi-supervised meta-learning approach use less performative methods as feature extractors (CACTUs: BiGAN, ACAI/DC; UMTRA: a simple, 4-layer CNN). An ablation study that ablates the use of SimCLR with Meta-GMVAE is necessary to ascertain whether the improvement is due to using SimCLR vs. using components attributable to Meta-GMVAE.

3) **Originality**: Highly relevant work on GMM priors for VAEs is not cited in the submission: [4, 5]. The submission also does not discuss variations on the VAE that address the meta-learning setting (e.g., [6, 7]), which also demonstrate how the VAE formulation in (1) derives from a hierarchical model (cf. the non-hierarchical model on which the original VAE formulation is based).

##### Minor points:

There are errors in reproducing the results from [1] in Table 2 of the submission (some percentages are incorrect); these errors do not affect the ranking of comparisons.

##### References:

[1] [Hsu, Kyle, Sergey Levine, and Chelsea Finn. "Unsupervised learning via meta-learning." In ICLR, 2019.](https://arxiv.org/abs/1810.02334)

[2] [Vinyals, Oriol, Charles Blundell, Timothy Lillicrap, and Daan Wierstra. "Matching networks for one-shot learning." In Advances in neural information processing systems, pp. 3630-3638. 2016.](http://papers.nips.cc/paper/6385-matching-networks-for-one-shot-learning)

[3] [Ravi, Sachin, and Hugo Larochelle. "Optimization as a model for few-shot learning." In ICLR, 2017.](https://openreview.net/pdf?id=rJY0-Kcll)

[4] [Dilokthanakul, Nat, Pedro AM Mediano, Marta Garnelo, Matthew CH Lee, Hugh Salimbeni, Kai Arulkumaran, and Murray Shanahan. "Deep unsupervised clustering with gaussian mixture variational autoencoders." arXiv preprint arXiv:1611.02648 (2016).](https://arxiv.org/abs/1611.02648)

[5] [Jiang, Zhuxi, Yin Zheng, Huachun Tan, Bangsheng Tang, and Hanning Zhou. "Variational deep embedding: An unsupervised and generative approach to clustering." In IJCAI, 2017.](https://arxiv.org/abs/1611.05148)

[6] [Hewitt, Luke B., Maxwell I. Nye, Andreea Gane, Tommi Jaakkola, and Joshua B. Tenenbaum. "The variational homoencoder: Learning to learn high capacity generative models from few examples." In UAI, 2018.](https://arxiv.org/abs/1807.08919)

[7] [Garnelo, Marta, Jonathan Schwarz, Dan Rosenbaum, Fabio Viola, Danilo J. Rezende, S. M. Eslami, and Yee Whye Teh. "Neural processes." In ICML, 2018.](https://arxiv.org/abs/1807.01622)

[8] [Triantafillou, Eleni, Tyler Zhu, Vincent Dumoulin, Pascal Lamblin, Utku Evci, Kelvin Xu, Ross Goroshin et al. "Meta-dataset: A dataset of datasets for learning to learn from few examples." In ICML, 2020.](https://arxiv.org/abs/1903.03096)

---

> ### Author Response · Authors · 2020-11-13
> **Regarding 3. Originality**
>
> **Originality:** The references to some very relevant work is missing.
>
> Thank you for the helpful suggestion. We have include detailed discussions about the suggested references in the related work section of the **revision**. We discuss them in comparison to our method below:
>
> (3.1) Highly relevant work on GMM priors for VAEs is not cited in the submission: [4, 5]
>
> - The assumption of using a GMM prior distribution from [4, 5] is similar to ours. However, our framework is fundamentally different from [4] and [5] since ours is an **unsupervised meta-learning**framework which aims to learn a dataset-conditioned GMM prior over a large number of tasks, such that it can estimate the GMM prior on an **unseen task**(dataset). [4] and [5] are single-task learning methods and their prior distributions **cannot adapt**to the new task, as they are fixed. We will cite and include the above discussion in the revision.
>
> ---
>
> (3.2) The submission also does not discuss variations on the VAE that address the meta-learning setting (e.g., [6, 7]).
>
> - Both [6] and [7] are relevant to our work in that they model the marginal log-likelihood of a dataset using the VAE framework, and utilize a set-level encoding. However, note that they are **supervised learning**approaches that models the **unimodal** variational **posterior**distributions of the latent variables using class information, and thus are fundamentally different from ours. We have discussed about the following differences to [6] and [7] in the revision.
>
> > a) First of all, [6] and [7] **do not consider unsupervised meta-learning**. In their settings, each dataset consists of data points from the same concept (e.g. class). Contrarily, our method assumes that the data points are unlabeled.
>
> > b) Moreover, both [6] and [7] model the set-dependent variational posterior with a **single global latent variable**. In contrast, we model the variational posterior conditioned on each data instance. Note that we model the GMM prior to be conditioned on the dataset, and not the posterior.
>
> ---
>
> Minor points: There are errors in reproducing the results from [1] in Table 2 of the submission (some percentages are incorrect); these errors do not affect the ranking of comparisons.
> - Thank you for pointing them out. We have revised the errors in the revision.

---

> > ### Comment · AnonReviewer3 · 2020-11-23
> > **Thanks for incoporating those references.**
> >
> > See my above response for a comment on how to improve clarity in the new additions.

---

> ### Author Response · Authors · 2020-11-13
> **Regarding 2. Quality**
>
> We provide response to your comments regarding quality below:
>
> (2.1) The experimental evaluation does not provide a measure of variance.
> - We have included the standard errors for 95% confidence interval of both datasets **in the revision**(Section B.1 and Section C.1) as follows:
>
> |    Omniglot     | 5-way, 1-shot | 5-way, 5-shot | 20-way, 1-shot | 20-way, 5-shot |
> |:--------------|:-------------:|:-------------:|:--------------:|:--------------:|
> |   Meta-GMVAE   |  94.92 ± 0.42 |  97.09 ± 0.20 |  82.21 ± 0.44  |  90.61 ± 0.19  |
>
> |    Mini-ImageNet     | 5-way, 1-shot | 5-way, 5-shot | 5-way, 20-shot | 5-way, 50-shot |
> |:--------------|:-------------:|:-------------:|:--------------:|:--------------:|
> |   Meta-GMVAE   |  42.82 ± 0.56 |  55.73 ± 0.48 |  63.14 ± 0.47  |  68.26 ± 0.42  |
>
> ---
>
> (2.2) The comparison to baselines is not fair due to the use of SimCLR.
> - First note that we do not use SimCLR for Omniglot experiments. Moreover, please note that we are using a similar architecture used by the baselines, **not the ResNet-50**used in the original SimCLR paper. Thus our method does not benefit from any unfair advantages (e.g. no additional labels or significantly increased model capacity) over the baselines. The reason we use SimCLR is because we wanted to show that for complex input images such as Mini-ImageNet on which VAE may not be accurate, we do not need to generate the input images but could generate high-level visual features instead. Moreover, the **ablation study (in the Table 2-Left)** shows that each component of Meta-GMVAE significantly improves the performance over the base model pre-trained with SimCLR.

---

> > ### Comment · AnonReviewer3 · 2020-11-23
> > **Comparison wrt baselines is still not entirely fair due to the use of SimCLR; ablation study seems to be missing a case**
> >
> > (2.2, SimCLR) If SimCLR does not provide an "unfair advantage", then why not use the representation learning methods employed by the baselines (BiGAN, ACAI/DC) instead of SimCLR, or update the baselines to make use of SimCLR? This is the more proper comparison. Moreover, it is contradictory to simultaneously claim that SimCLR does not provide an advantage but that it is a "powerful unsupervised representation learning approach," which implies it benefits the method.
> >
> > (2.2, ablation) The ablation study is a bit confusing, of the use of the label "Meta-GMM" instead of a categorical label for each component that is ablated/added in, so I'd recommend changing the format. I cannot determine the variant that ablates the use of SimCLR properly. It cannot be "Vanilla VAE + EM" (no SimCLR--VAE instead) vs. "Meta-GMM w/o Set, σ2" (yes SimCLR), because the Omniglot numbers improve (and Omniglot does not use SimCLR); in particular, there must be another change when going from "Vanilla VAE + EM" to "Meta-GMM w/o Set, σ2".
> >
> > To be clear: I am attempting to understand the performance due to using Meta-GMM (vs. another unsupervised meta-learning method), controlling for the benefit due to using SimCLR.

---

> > > ### Author Response · Authors · 2020-11-23
> > > **Regarding 2. Quality (2)**
> > >
> > > We sincerely appreciate your helpful suggestions. Including the new experimental results as well as the new formatting of the ablation study you suggested have significantly improved the quality of our paper. We respond to your comments as below:
> > >
> > > ---
> > >
> > > **1.** If SimCLR does not provide an "unfair advantage", then why not use the representation learning methods employed by the baselines (BiGAN, ACAI/DC) instead of SimCLR, or update the baselines to make use of SimCLR?
> > >
> > > - Based on the public code provided by [CACTUs's authors](https://github.com/kylehkhsu/cactus-maml) and [UMTRA's authors](https://github.com/siavash-khodadadeh/UMTRA-Release), we ran experiment on the baselines with SimCLR. For CACTUs, we generate pseudo-classes by performing clustering in the embedding space pretrained by SimCLR. For UMTRA, we follow the exact same procedure to generate training episode, which is proposed by authors.  For a fair comparison of all methods, we utilize the same SimCLR features we used in our Mini-ImageNet experiments. The table below shows that using SimCLR helps the baselines  (CACTUs, UMTRA) obtain improved performance on the some settings (5-way 1-shot) but not with larger shots, compared with the models in the original papers. Thus the new results do not change any message from the original paper. We have included this comparison in Table 7 of the Appendix C.2.
> > >
> > > |    Mini-ImageNet     | 5-way, 1-shot | 5-way, 5-shot | 5-way, 20-shot | 5-way, 50-shot |
> > > |:--------------|:-------------:|:-------------:|:--------------:|:--------------:|
> > > |   CACTUs-MAML (DC)   |  39.90 |  53.98 |  63.84  |  69.64  |
> > > |   CACTUs-MAML (SimCLR)   |  40.39 |  52.35 |  61.09  |  64.89  |
> > > |   UMTRA   |  39.93 |  50.73 |  61.11  |  67.15  |
> > > |   UMTRA (SimCLR)   |  40.85 | 51.47 |  61.03  |  67.30  |
> > > |   Meta-GMVAE  |  42.82 |  55.73 |  63.14  |  68.26  |
> > >
> > > ---
> > >
> > > **2.** The ablation study is a bit confusing, of the use of the label "Meta-GMM" instead of a categorical label for each component that is ablated/added in, so I'd recommend changing the format.
> > >
> > > - Thank you for your helpful suggestion. We have included a categorical label for each component (i.e., $\checkmark$ with Ep, Set, and $\sigma^2$ columns) in the revision.
> > >
> > > ---
> > >
> > > **3.** It cannot be "Vanilla VAE + EM" (no SimCLR--VAE instead) vs. "Meta-GMM w/o Set, σ2" (yes SimCLR), because the Omniglot numbers improve (and Omniglot does not use SimCLR); in particular, there must be another change when going from "Vanilla VAE + EM" to "Meta-GMM w/o Set, σ2".
> > >
> > > - We use SimCLR pretrained features for **all the Mini-ImageNet experiments including the "Vanilla VAE + EM" baseline**.  Vanilla VAE + EM" (which utilizes SimCLR features) even underperforms "SimCLR + LogisticRegression", which shows that a simple combination of SimCLR  and "Vanilla VAE + EM" does not improve the performance. Thus this is a result that clearly shows the advantage of our proposed Meta-GMM. We have clarified that we use SimCLR pretrained feature for "Vanilla VAE + EM" variants by modifying it into "Vanilla VAE (SimCLR)" in Table 2 (Left) of the revision.

---

> > > > ### Comment · AnonReviewer3 · 2020-11-24
> > > > **Thanks for the additional comparisons & ablations.**
> > > >
> > > > 1. Great; thank you for adding these fairer baselines.
> > > >
> > > > 2. Looks good. I'd also suggest treating "SimCLR features" as a categorical label as well, to prevent confusions such as mine.
> > > >
> > > > 3. Thanks for the clarification. It's necessary to include "2) Vanilla VAE" and "Meta-GMVAE (no SimCLR)" for the miniImageNet dataset to precisely quantify the improvement due to using SimCLR features, or explain why the use of SimCLR features is necessary (e.g., for convergence) on this dataset.

---

> > > > > ### Author Response · Authors · 2020-11-24
> > > > > **Answer**
> > > > >
> > > > > **Great; thank you for adding these fairer baselines.**
> > > > > - We sincerely thank you for your helpful suggestion that made the baselines fairer.
> > > > >
> > > > > **Looks good. I'd also suggest treating "SimCLR features" as a categorical label as well, to prevent confusions such as mine.**
> > > > >
> > > > > - We tried to include SimCLR as a categorical label in Table 2(Left), but then it made the Table look awkward since the Omniglot dataset did not use SimCLR. We will find out a way to format the Table as you suggest, to avoid any confusion.
> > > > >
> > > > > **It's necessary to include "2) Vanilla VAE" and "Meta-GMVAE (no SimCLR)" for the miniImageNet dataset to precisely quantify the improvement due to using SimCLR features, or explain why the use of SimCLR features is necessary (e.g., for convergence) on this dataset.**
> > > > >
> > > > > - The main reason we use SimCLR for miniImageNet experiments, is to show that we could use the model to reconstruct **high-level features**instead of **input samples**, since generating the actual samples (e.g. images) may not be necessary for capturing the meta-knowledge for non-generative downstream tasks (e.g. few-shot classification, Page 2, 2nd paragraph).
> > > > > - Also, as mentioned in the last paragraph of Page 6, a vanilla VAE does not train well with complex images such as images from MiniImageNet. Larger architectures such as NVAE [Vahdat and Kautz 20] may successfully learn to generate them, but we find it unnecessary since our goal is to learn meaningful features, not on generating actual image samples.
> > > > > - We will include the Vanilla VAE and Meta-GMVAE (no SimCLR) results in the ablation study table in the revision, as you suggested. If we cannot finish it by the end of the rebuttal deadline, we will include it in the final version of the paper, if accepted.
> > > > >
> > > > > Please let us know if there is anything else we should address, since the interactive discussion period will end in less than 19 hours.
> > > > >
> > > > > [Vahdat and Kautz 20] NVAE: A Deep Hierarchical Variational Autoencoder, NeurIPS 2020

---

> > > > > > ### Comment · AnonReviewer3 · 2020-11-24
> > > > > > **Concerns have been addressed; please do add the requested ablation to a future revision.**
> > > > > >
> > > > > > > The main reason we use SimCLR for miniImageNet experiments, is to show that we could use the model to reconstruct high-level features instead of input samples
> > > > > >
> > > > > > This is the same reasoning that prior works used to motivate the use of unsupervised feature extractors (i.e., "dimensionality reduction"). My main concern here is that, without the proper ablations, it is impossible to determine if using SimCLR in particular confers an improvement that is independent of the proposed method.
> > > > > >
> > > > > > > We will include the Vanilla VAE and Meta-GMVAE (no SimCLR) results in the ablation study table in the revision, as you suggested. If we cannot finish it by the end of the rebuttal deadline, we will include it in the final version of the paper, if accepted.
> > > > > >
> > > > > > The results of the SimCLR ablation on miniImageNet should be included in a future revision; however, I do not need to see them at this time since the rebuttal period is almost over. I would also recommend trying out one of the feature extractors used in prior work on unsupervised meta-learning (e.g., BiGAN) to understand the impact of input feature origin on your proposed method.
> > > > > >
> > > > > > > Please let us know if there is anything else we should address, since the interactive discussion period will end in less than 19 hours.
> > > > > >
> > > > > > I have no more questions.

---

> > > ### Author Response · Authors · 2020-11-24
> > > **Responses and the revision uploaded**
> > >
> > > Dear reviewer,
> > >
> > > We have uploaded the responses and the revision which faithfully reflects them. We revised inaccurate descriptions in the previous version of the paper for improved clarity, following your suggestions, and have performed experiments of the baselines with SimCLR, for a fair comparison with our work. The SimCLR experiments show that although using SimCLR for the baselines yields marginal performance gain on 1-shot cases, but not on others, and that our method still significantly outperform them under such a fair setting. We strongly believe that the improved clarity and additional experimental results will further strengthen our paper. Thank you for your helpful suggestions, for further improving the quality of our paper.

---

> > > > ### Comment · AnonReviewer3 · 2020-11-25
> > > > **Thanks for the revision; score updated.**
> > > >
> > > > Thanks for making those revisions; the paper has improved a lot. I have updated my score to reflect this.
> > > >
> > > > Some brief recommendations for future revisions:
> > > > - Section 3.2 is quite dense; I'd encourage you to break it up into smaller components, condense the writing and include the algorithm box in the main text.
> > > > - The introduction also can be condensed to be no more than a page.
> > > > - Move Table 3 to the appendix.

---

> > > > > ### Author Response · Authors · 2020-11-25
> > > > > **Thank you, we have revised the paper as you suggested**
> > > > >
> > > > > Dear reviewer,
> > > > >
> > > > > We sincerely appreciate you for your constructive comments, as well as responsiveness during the interactive discussion phase. It was indeed very helpful and we believe that our paper has become significantly more stronger, with more clear descriptions of the methods, improved presentation of the results, and fairer comparison against baselines.
> > > > >
> > > > > We have reflected all your suggestions in the revision.
> > > > > - **1.** We broke Section 3.2 into smaller components.
> > > > > - **2.** We included the algorithm box into the main text.
> > > > > - **3.** We condensed the introduction to be no more than two pages.
> > > > > - **4.** We moved Table 3 to the appendix.
> > > > >
> > > > > Thank you and we hope you stay safe.
> > > > >
> > > > > Best regards,
> > > > > Authors

---

> ### Author Response · Authors · 2020-11-13
> **Regarding 1. Clarity**
>
> **Clarity:** the algorithmic components of the submission were very difficult to get straight.
>
> - We sincerely appreciate your constructive comments. We have reflected your comments in the revision to improve the clarity of the paper. Please refer to the below for detailed discussions.
>
> (1.1) The VAE formulation is introduced without precedent just above equation (1).
>
> - We apologize for your confusion. We derived the VAE formulation with the i.i.d assumption that the log likelihood of a dataset at each episode equals the sum over the log-likelihoods of the data points belonging to the dataset, as follows:
> $\log p_\theta(\mathcal{D_i}) = \sum_{j=1}^{M} \log p_\theta (x_j) = \sum_{j=1}^{M} \log \int p_\theta (x_j|z) p_{\psi_i}(z) \frac{q_\phi(z|x_j, \mathcal{D}_i)}{q_\phi(z|x_j, \mathcal{D}_i)} dz$.
> We have **included** this precedent derivation in the **revision**.
>
> ---
>
> (1.2) This is the first time an "isotropic Gaussian distribution" is mentioned in the method.
>
> - For improved clarity, we have **revised**the sentence as follows: "Following the convention of the VAE literature, we assume that the variational posterior follows an isotropic Gaussian distribution. ... Thus we learn a set-conditioned variational posterior $q_\phi(z|x, \mathcal{D_i})$, which models a latent variable to encode each data $x_j$ within the given dataset $\mathcal{D_i}$ into the latent space."
>
> ---
>
> (1.3) self-attention is not explained further, and there is no explanation of how the convolutional neural network (CNN) fits into the whole framework.
>
> - We have omitted it in the submission due to the page limit. To model the variational posterior $q_\phi(z|x, \mathcal{D_i})$, we use the self-attention mechansim as follows:
> $H = \text{TransformerEncoderLayer}( f(\mathcal{D_i}) )$, $\mu(x_j)  = W_{\mu} H_j + b_{\mu},$ and $\sigma^2(x_j) = exp(W_{\sigma^2} H_j + b_{\sigma^2})$,
> where $\text{TransformerEncoder}(\cdot)$ is a neural network based on the multi-head self-attention mechanism proposed by [Vaswani et al, NeuRIPS 2017], and $f(\cdot)$ can be either a convolutional neural network (for Omniglot) or an identity function (Mini-ImageNet).  $W_{\mu}$ and $W_{\sigma^2}$ are projection weight matrices, $ b_{\mu}$ and $b_{\sigma^2}$ are bias vectors, and $exp$ is an element-wise exponential function. We have **included**this in the **revision**.
>
> ---
>
> (1.4)  $y$ and $z$ are not yet defined except by reference to the VAE formulation in (1), but that was insufficiently explained as a part of the algorithm.
>
> - In the revision, we have included the following **generative process**of Meta-GMVAE, to further clarify how the latent variable does work in our framework:
> - a) $y \sim p_{\psi_i^*}(y)$, where $y$ is a label of a data instance and ${\psi_i^*}$ is the optimal prior parameter for the dataset $\mathcal{D_i}$ given at the episode $i$.
> - b) $z \sim p_{\psi_i^*}(z|y)$, where $z$ corresponds to the Gaussian latent variable responsible for data generation.
> - c) $x \sim p_{\theta}(x|z)$, where $\theta$ is the parameter of the generative model.
> - As with conventional VAEs, we use a variational posterior parameterized by $\phi$ to approximate the true posterior.
>
> ---
>
> (1.5) how the GMM parameters are initialized for EM.
> - The details regarding GMM initialization is already **described in the paragraphs below the equations (5) and (8)** (equation (7) and (10) in the revision), however, we have included the **pseudo-code**of the algorithm in the revision for more details.
>
> ---
>
> (1.6) While component architectures and experimental setups are detailed in the appendix, how everything fits together is not adequately described.
> - We have included the **pseudo-code**of the overall algorithm (Algorithm 1 and 2) in the Section A of the Appendix, in the revision.

---

> > ### Comment · AnonReviewer3 · 2020-11-23
> > **The additions improve clarity, so clarity is no longer a primary concern, but some things remain to be cleaned up**
> >
> > Thanks for making the clarity improvements--many of the missing details have been added now. The algorithm box is appreciated.
> >
> > However, I do think that the paper could be improved by adding more precision to the comparisons. For example:
> >
> > - "Our derivation of the evidence lower bound in Eq 4 is somewhat similar..." How, precisely (i.e., in reference to specific equations / variables), is it somewhat similar?
> > - "target single-task learning and the parameter of the prior network is fixed after training" Explain, precisely (i.e., in reference to specific equations / variables), how your method does not fix these parameters after training.
> >
> > I'd also recommend in the text (i.e., not just in the algorithm) more clearly making the distinction between the episode dataset phase and the full dataset.
> >
> > Also, I disagree with one of the points made wrt prior work:
> >
> > -  "...previous works assume that each input set consists of data instances from a single concept (e.g. a class), therefore, they encode the dataset into a single global latent variable (e.g. qφ(z|D)). However, this is not possible for unsupervised meta-learning where labels are unavailable. " That it is not possible to encode a dataset of images *without labels* into a single latent representation is false--it is *entirely* possible to represent a labelless dataset in this way. In fact, your task-specific parameter $\psi$ is such a representation.

---

> > > ### Author Response · Authors · 2020-11-23
> > > **Regarding 1. Clarity (2)**
> > >
> > > Thank you for your suggestion to improve clarity of our paper. We respond each questions as below:
> > >
> > > ---
> > >
> > > **1.** "Our derivation of the evidence lower bound in Eq 4 is somewhat similar..." How, precisely (i.e., in reference to specific equations / variables), is it somewhat similar?
> > >
> > > - As mentioned before in (1.1), we derived the VAE formulation with the i.i.d assumption that the log likelihood of a dataset at each episode equals the sum over the log-likelihoods of each individual data point. This is similar to the previous derivations, **such as Equation 3 in Neural Statistician [1] and Equation 4 in Variational Homoencoder [2]**. We have included this description in the revision.
> > >
> > > ---
> > >
> > > **2.** "target single-task learning and the parameter of the prior network is fixed after training" Explain, precisely (i.e., in reference to specific equations / variables), how your method does not fix these parameters after training.
> > >
> > > - The parameter of the prior network is fixed after training, for example, **Equation 1c in [3] and equation 5 in [4]**, and is agnostic of the dataset given at the test time. In our case, the parameter of GMM prior $\psi$ can flexibly adapt to a novel dataset for a given episode using the EM algorithm as suggested in Equation 6 of our paper. We have included this description in the revision.
> > >
> > > ---
> > >
> > > **3.** "...previous works assume that each input set consists of data instances from a single concept (e.g. a class), therefore, they encode the dataset into a single global latent variable (e.g. $q_\phi(z|\mathcal{D})$). However, this is not possible for unsupervised meta-learning where labels are unavailable. " That it is not possible to encode a dataset of images without labels into a single latent representation is false--it is entirely possible to represent a labelless dataset in this way. In fact, your task-specific parameter $\psi$  is such a representation.
> > >
> > > - We did not exactly say nor meant that it is not possible to encode dataset of images without labels into a single representation, but rather saying that incorporating label information for unsupervised meta-learning is not possible. However, since this is trivially true, we **have removed the part "This is not possible"** to avoid any confusion in the revision. As you mentioned, it is possible to represent a label-less dataset using a single global latent variable, but was never done since they were considering labeled data. Even when the label information is not directly used, they are implicitly used, as the dataset belonging to a specific class was given to the model.
> > >
> > > ---
> > >
> > > References:
> > >
> > > [1] [Harrison Edwards, and Amos Storkey. "Towards a Neural Statistician." In ICLR, 2017.](https://arxiv.org/pdf/1606.02185.pdf)
> > >
> > > [2] [Luke B. Hewitt et al. "The Variational Homoencoder: Learning to learn high capacity generative models from few examples." In UAI, 2018.](https://arxiv.org/pdf/1807.08919.pdf)
> > >
> > > [3] [Dilokthanakul et al. "Deep Unsupervised Clustering with Gaussian Mixture Variational Autoencoders." In arXiv, 2016](https://arxiv.org/abs/1611.02648)
> > >
> > > [4] [Zhuxi Jiang et al. "Variational Deep Embedding: An Unsupervised and Generative Approach to Clustering." In IJCAI, 2017](https://arxiv.org/pdf/1611.05148.pdf)

---

> ### Author Response · Authors · 2020-11-23
> **Responses and the revision uploaded**
>
> Dear Reviewer,
>
> Could you please go over our responses and the revision since we can have interactions with you only by this Tuesday (24th)? We have responded to your comments and faithfully reflected them in the revision. We sincerely thank you for your time and efforts in reviewing our paper, and your insightful and constructive comments.
>
> Thanks, Authors

---

### Official Review · AnonReviewer2 · 2020-11-09
**Interesting method for unsupervised meta-learning**

**Rating:** 7
**Confidence:** 4

**Review:**

This paper proposes a method for unsupervised meta-learning based on using a variational autoencoder (VAE). The variational autoencoder model they use differs from the typical one in that it considers episode-specific datasets, where the approximate posterior can be computed as a function of the set (using transformer architecture) rather than using an individual example. Additionally, they use a mixture of Gaussian distribution as a prior, whose parameters are learned per-episode using the EM algorithm. For the supervised evaluation phase, in order to adapt the learned prior to the few-shot dataset setting, semi-supervised EM is run using both support and query sets to adapt the mixture of Gaussian distribution to the evaluation dataset. Then, the query set predictions are obtained using the learned prior and posterior from the VAE model. Experimental evaluation is conducted on the Omniglot and Mini-ImageNet benchmarks and the proposed method is compared against other unsupervised meta-learning methods, mainly CACTUs and UMTRA. An interesting aspect about the Mini-ImageNet experiments are that because learning the VAE directly for this high-dimensional data may be difficult, the authors use features from a SimCLR-trained model as input for their VAE model. The proposed method seems to perform favorably across both of the benchmarks when varying the number of "shots".

Pros
* Whereas previous work in unsupervised meta-learning involved creating unsupervised episodes for meta-training (via augmentations or clustering of unsupervised model features), this paper takes a very different route but still seems to achieve very good performance.
* The authors were able to scale their model to the Mini-ImageNet dataset by using SimCLR-trained features and with this choice, the final model attains good performance on the benchmark compared to previous work.

Cons
* This is a not necessarily a big con but a point that could be clarified further. How are the number of components for the GMM decided for meta-training? How does the choice of the number of components impact how the GMM is used at evaluation-time? Would it not pose an issue that during training we may have more/less components than are actually necessary at evaluation-time depending on the number of classes we are considering at evaluation-time? Is it the case that a separate model needs to be trained if number of evaluation classes is changed i.e. from 1-shot, 5-class to 1-shot, 10-class?
* I believe the paper could be improved by adding an algorithm description of how exactly the model is trained and how evaluation takes place in terms of exact steps The algorithmic pseudocode can reference equations within the paper but I believe this would greatly help in terms of understanding how to recreate the exact training and evaluation procedure for the proposed model.

---

> ### Author Response · Authors · 2020-11-13
> **Response to R2**
>
> We really  appreciate your constructive comments. We respond to the individual comments below:
>
> **1.** How can we decide the number of components for Meta-GMVAE for meta-training?
> - In case if we know the number of classes $K$ for meta-test, we can set the number of component as $K$. However, we can also simply set the number of components to an **arbitrary number**, since our method can **generalize to variable-way classification tasks**at meta-test time (Please see the **Table 2 (right)**). It may be also possible to treat the number of classes as a hyperparameter and find an optimal value using a cross-validation if the target task is fixed.
>
> ---
>
> **2.** How does the choice of the number of components impact how the GMM is used at evaluation-time?
> - As shown in the **Table 2(right)**, the performance loss of Meta-GMVAE when there is a mismatch in the number of components across meta-training and meta-test is **negligible**(e.g. **81.98 vs. 81.11**on 20-way classification at meta-test, with 20-way 5-shot and 5-way 20-shot training respectively). This is another important advantage of our unsupervised meta-learning method since it can generalize to tasks with varying number of ways at meta-test time.
>
> ---
>
> **3.** Would it not pose an issue that during training we may have more/less components than are actually necessary at evaluation-time depending on the number of classes we are considering at evaluation-time?
> - As discussed in the answer to the previous questions, and shown in Table 2-Right, this is not an issue for our model since our model can adapt to classification problems with varying number of classes. For example, when we train the model with 5 components (K=5) for 10-way few-shot classification, we can just perform semi-supervised EM by setting K to 10. Moreover, as shown in **Table 2 (Right)**, it **does not harm**the performance of our Meta-GMVAE on tasks with any number of ways.
>
> ---
>
> **4.**  Is it the case that a separate model needs to be trained if number of evaluation classes is changed?
> - As shown in **Table 2 (right)**, Meta-GMVAE consistently achieves the best performance on classification tasks with any number of ways (classes) with negligible performance degeneration when there is a large mismatch between the number of ways for meta-training and meta-test.  Thus, we believe that we do not need to train a separate model for each case, and this is a **main advantage**of using our unsupervised meta-learning method over supervised methods, as ours can generalize to diverse downstream tasks.
>
> ---
>
> **5.** The paper could be improved by adding an algorithm description.
> - Thank you for your helpful suggestion. We have provided the algorithm in Section A of Appendix A (Algorithm1, Algorithm2).

---

### Author Response · Authors · 2020-11-15
**Pseudo-code of the algorithm for Meta-training and Meta-test**

We provide the pseudo-code of the algorithm for meta-training and meta-test requested by R2, R3 and R4 here. We also included them in the Algorithm 1 and 2 in the Section A of the Appendix.

---

**Algorithm1:** Meta-training of Meta-GMVAE

**Require**: An unlabeled dataset $\mathcal{D}_u$

1:&nbsp;Initialize parameters $\theta$ and $\phi$

2:&nbsp;**while** not done **do**

3:&nbsp;&nbsp;&nbsp;&nbsp;&nbsp;Sample $B$ episode datasets {$\mathcal{D}_i$}$_i^B$ from $\mathcal{D}_u$

4:&nbsp;&nbsp;&nbsp;&nbsp;&nbsp;**for all** $i \in [1,B]$ **do**

5:&nbsp;&nbsp;&nbsp;&nbsp;&nbsp;&nbsp;&nbsp;&nbsp;&nbsp;Draw $N$ MC samples from $q_\phi(z_j|x_j, \mathcal{D}_i)$

6:&nbsp;&nbsp;&nbsp;&nbsp;&nbsp;&nbsp;&nbsp;&nbsp;&nbsp;Initialize the parameter $\pi_k$ as $\frac{1}{K}$, and $\mu_k$ with randomly drawn $K$ points

7:&nbsp;&nbsp;&nbsp;&nbsp;&nbsp;&nbsp;&nbsp;&nbsp;&nbsp;Compute optimal parameter $\psi_i^*$ using EM algorithm in Eq 7

8:&nbsp;&nbsp;&nbsp;&nbsp;&nbsp;**end for**

9:&nbsp;&nbsp;&nbsp;&nbsp;&nbsp;Evaluate $\mathcal{L}(\theta, \phi,$ {$\mathcal{D}_i$}$_i^B$) using the lowerbound in Eq 9

10:&nbsp;&nbsp;&nbsp;&nbsp;&nbsp;Update $\theta$ and $\phi$ by taking gradient ascent of $\mathcal{L}(\theta, \phi,$ {$\mathcal{D}_i$}$_i^B$)

---

**Algorithm2:** Meta-test of Meta-GMVAE for a single episode

**Require**: A test task $\mathcal{T}=\mathcal{S}\cup\mathcal{Q}$

1:&nbsp;Set $\mathcal{D}$ as all $x \in \mathcal{T}$

2:&nbsp;Draw $N$ MC samples from $q_\phi(z_j|x_j, \mathcal{D})$

3:&nbsp;Initialize $\mu_k= \frac{\sum_{s,n=1}^{S,N} 1_{y_s^{(n)}=k} z_s^{(n)}}{ \sum_{s,n=1}^{S,N} 1_{y_s^{(n)}=k}} $ and $\sigma^2=I$

4:&nbsp;Compute optimal parameter $\psi^*$ using semi-supervised EM algorithm in Eq 10

5:&nbsp;Compute the conditional probability $p(y_q|x_q, \mathcal{D})$ using Eq 11

6:&nbsp;Infer the label $y_q=\underset{k}{\arg\max}p(y_q=k|x_q, \mathcal{D})$

---

---

### Author Response · Authors · 2020-11-18
**Summary of the Revision**

We really appreciate all the reviewers for their constructive comments. Here we briefly summarize the summary of the revision:

- We have included the discussions on the suggested related works that **use the Gaussian mixture prior for VAE and learn the VAE posterior in a set-dependent manner**, in the 2. Related Work section and 3.2 Meta-Level Gaussian Mixture VAE section.

- In order to improve clarity, we have included detailed descriptions of **the generative process and the implementation of the variational posterior** in the 3.2 Meta-Level Gaussian Mixture VAE section.

- We have included **the pseudo-code of the algorithm**in Section A of the Appendix (Algorithm 1 and 2)

- We included the results with the **95% confidence intervals** in Section B.1 (Omniglot) and Section C.1 (MiniImageNet) of the Appendix.

---

- Based on the request from R1, we have clarified our contributions in the **two bullet points** at the end of the Introduction section.

---

### Comment · ~D._Khuê_Lê-Huu1 · 2021-04-22
**Ideas seem to be very similar to "Continual Unsupervised Representation Learning" by Rao et al. (NeurIPS 2019)**

Hello,
First of all congratulations on the acceptance!
I would like to point out a paper published by Rao et al. two years ago at NeurIPS 2019 that proposed very similar ideas: "Continual Unsupervised Representation Learning". In general there is a lot of overlap between meta-learning and continual learning (in unsupervised settings), in the sense that a method for solving one could be applied to solving the other in a straightforward manner. I am surprised that none of the reviewers were aware of this work.
Would you have some comments on the difference between your work and Rao et al.?
Thanks.

---

> ### Comment · AnonReviewer4 · 2021-04-23
> **Not that similar at all**
>
> Before the authors chime in, I'd like to clarify, that at least some of the reviewers were aware of this work, but the similarities don't seem to be too strong in my opinion:
> 1. CURL solves a continual-learning problem, whereas Meta-GMVAE a meta-learning one. It would be useful to clarify what do you mean by "one could be applied to solving the other in a straightforward manner".
> 2. A big part of CURL is dynamical expansion of the clusters, whereas Meta-GMVAE (as it's solving a different problem, with a known number of classes) can assume the fixed number of groups.
> 3. Meta-GMVAE relies on subsampling the dataset into pieces, whereas CURL switches between using (regularly sampled) data from the dataset and the synthetic one from the generative model.

---

> > ### Author Response · Authors · 2021-04-23
> > **We agree with R4's opinion**
> >
> > We sincerely appreciate for explaining the difference between Meta-GMVAE and CURL. Thank you so much!

---

> > ### Comment · ~D._Khuê_Lê-Huu1 · 2021-05-01
> > **Thank R4 for the clarifications**
> >
> > Thanks, R4, for the clarifications. Please see my joint response to the authors above.

---

> ### Author Response · Authors · 2021-04-23
> **Response to your question**
>
> Although CURL and Meta-GMVAE are distantly relevant in the sense that they both use Gaussian mixtures to learn representations in an unsupervised manner, the two models are largely different in their goals, capabilities in adapting to new tasks, as well as in their technical components. We list their differences below:
>
> CURL
> - uses the model that generates Gaussian mixture to remember the representations of the instances from the past tasks (Equation 6 in [1]), and has **no mechanism** to adapt to an unseen task without training.
> - does not explicitly consider tasks and everything is modeled as sample-dependent.
> - uses a neural network to perform amortized inference of the instance- and label-dependent means, variances, and the mixing coefficients of the Gaussian mixture (Equation 1 in [1]).
>
> Meta-GMVAE
> - aims to adapt to **unseen tasks without training**, and has an explicit mechanism to obtain a task-specific model on unseen tasks for improved generalization.
> - explicitly models task (dataset)-dependent posterior and the prior,
> - uses a set-encoding to infer the task-dependent posterior (unimodal Gaussian) as in Equation 5.
> - uses the EM algorithm to infer the task-dependent prior (a mixture of Gaussians) as in Equation 7 and 10.
>
> In sum, except that the both methods use Gaussian Mixtures, CURL is irrelevant to our goal of fast adaptation to novel tasks, since its goal is mainly remembering (or being able to generate) past instances to combat catastrophic forgetting in continual learning. To be specific, our main novelty is in the proposal of a principled unsupervised meta-learning, which utilizes set-based amortized inference and EM algorithm to infer the tasks-dependent prior for the fast-adaptation to new tasks.
>
> It is clear from the reviews that the reviewers are fully aware of existing works (such as [2, 3, 4, 5] suggested by R4) that either learn or utilize mixtures of Gaussians for unsupervised representation learning similarly as CURL does. Reading through the paper and our responses in detail will make the differences between the two more obvious.
>
> [1] [Dushyant Rao, Francesco Visin, Andrei A. Rusu, Yee Whye Teh, Razvan Pascanu, and Raia Hadsell. "Continual Unsupervised Representation Learning." In NeuRIPS, 2019](https://arxiv.org/pdf/1910.14481.pdf)
>
> [2] [Nat Dilokthanakul, Pedro A. M. Mediano, Marta Garnelo, Matthew C. H. Lee, Hugh Salimbeni, Kai Arulkumaran, and Murray Shanahan. "Deep Unsupervised Clustering with Gaussian Mixture Variational Autoencoders." In arXiv, 2016.](https://arxiv.org/abs/1611.02648)
>
> [3] [Qingyu Zhao, Nicolas Honnorat, Ehsan Adeli, and Kilian M. Pohl. "Truncated Gaussian-Mixture Variational AutoEncoder." In arXiv, 2019](https://arxiv.org/abs/1902.03717)
>
> [4] Chunsheng Guo,  Jialuo Zhou, Huahua Chen, Na Ying, Jianwu Zhang, and Di Zhou. "Variational Autoencoder With Optimizing Gaussian Mixture Model Priors." In IEEE, 2019 (10.1109/ACCESS.2020.2977671.)
>
> [5] [Linxiao Yang, Ngai-Man Cheung, Jiaying Li, and Jun Fang, "Deep Clustering by Gaussian Mixture Variational Autoencoders with Graph Embedding." In ICCV, 2019](https://openaccess.thecvf.com/content_ICCV_2019/papers/Yang_Deep_Clustering_by_Gaussian_Mixture_Variational_Autoencoders_With_Graph_Embedding_ICCV_2019_paper.pdf)

---

> > ### Comment · ~D._Khuê_Lê-Huu1 · 2021-05-01
> > **Joint response to the authors and R4. I thank them for the clarifications but I still believe that Rao et al. merits a discussion.**
> >
> > First of all sorry for the late reply, I didn’t receive any notifications for your responses (not sure why but there doesn’t seem to have an option to enable that in my account settings either).
> >
> > I would like to thank the authors and R4 for their clarifications. I would also like to clarify, just in case, that I am not trying in any way to downgrade the merits of the paper. By contrast, I find the paper very interesting, its technical contributions significant, and its empirical results impressive. However, this does not mean that the paper cannot be further improved, and I am raising a potential issue for such improvements. In my previous message I used the term “very similar” that sounds a bit too strong, but I believe that Rao et al. shares some similarities with the current work and thus merits at least a brief discussion, as much as Dilokthanakul et al., 2016 and Jiang et al., 2017 do.
> >
> > Obviously when adapting an existing technique/model to a new task (in this case, from clustering in Dilokthanakul et al., 2016 and Jiang et al., 2017 to meta-learning in the current work, or to continual-learning in Rao et al.), a lot of technical details need to be addressed, resulting in novelty/differences. However, “there are differences” does not mean “there’s no similarity”. In this case, the similarities should be acknowledged and the differences should be clearly discussed. Let me recall that R1 and R3 also raised similar issues in their reviews. Indeed, they pointed out that the authors had missed the important references Dilokthanakul et al., 2016 and Jiang et al., 2017 (together with a discussion about these works) in their initial manuscript. (By the way, R4 has claimed to be aware of Rao et al. at the time of reviewing yet surprisingly R4 did not raise the same issue as R1 and R3 did, because Rao et al. made it clear in their paper that their generative process is the same as in Jiang et al., 2017, which is also the same as in the current work.) The authors correctly addressed R1 and R3’s concerns by adding the following important paragraph to the manuscript:
> >
> > "The above generative process is similar to those from the previous works (Dilokthanakul et al., 2016; Jiang et al., 2017) on modeling the VAE prior with Gaussian mixtures. However, they target single-task learning and the parameter of the prior network is fixed after training such as equation 1c in Dilokthanakul et al. (2016) and equation 5 in Jiang et al. (2017), which is suboptimal since a meta-learning model should be able to adapt and generalize to a novel task."
> >
> > In view of the above statement, the similarity with respect to Rao et al. is apparently more than just "both use Gaussian mixtures", because Rao et al. also "target multi-task learning and the parameter of the prior network is updated when learning new tasks" (compare this with the above statement). I mean, if the above is why the current work differs from Dilokthanakul et al., 2016 and Jiang et al., 2017, then it is also precisely why this work is similar to Rao et al.
> >
> > In case it’s not clear, consider the following scenario:
> >
> > Current work: two red balls.
> > Previous work A: one red ball.
> > Previous work B: two blue balls.
> >
> > The claim “The current work is different than previous work A because we have two red balls while they only have one” is technically correct, but is also problematic because it ignores previous work B for which the same argument doesn’t apply at all. (The claim may be fine for readers who are not aware of B, but for those who are, it doesn’t sound right.) It would be better to add something like “Ok, previous work B also have two balls as we do, but their balls are blue while ours are red”.
> >
> > I hope that my points are clear. I agree the with the differences that the authors and R4 have listed above (except maybe the “label-dependent versus task-dependent” statements from the authors, it should be noted that in Rao et al., “label” and “task” are the same thing, they used “label” as a synonym but it should be understood as “task”), but as I said, “there are differences” does not mean “there is no similarity”.

---

> > > ### Comment · ~Sung_Ju_Hwang1 · 2021-05-02
> > > **Brief clarification.**
> > >
> > > I appreciate your detailed feedback to our intial response. I would like to stress that the main technical novelty comes from the amortized unsupervised meta-learning framework that allows the model to instantly adapt the distribution of the representation to a new dataset (encoded with a set encoder), without having to train on it. This can be achieved whether the prior follows a mixture of Gaussian, a vamp prior, or a unimodal Gaussian. Also there is no learning of a global model that generate the mixture of Gaussian as in (Rao et al. 19), as our Gaussian mixture prior is obtained on the fly for each task using EM. There is **no learning of Gaussian mixtures at all**, and thus there is no ovelap between the two works in terms of techincal contributions. Although I believe that this is already clearly described in the paper, it may need more clarification.
> > >
> > > I do agree with you that Rao et al. 19 is somehow related to ours in an abstract level as unsupervised continual learning is loosely related to unsupervised meta-learning and that both use a mixture of Gaussians as a component, but disagree that it is more relevant to ours than other existing works on unsupervised meta-learning or set-amortized meta learning. Still, I appreciate your opinion and interests in our works, and will clarify this in the related work section of the revised camera-ready version of the paper.

---

> > > > ### Comment · ~D._Khuê_Lê-Huu1 · 2021-05-02
> > > > **Thanks for the clarifications**
> > > >
> > > > Thanks for your clarifications (and also for being very responsive, as always). This is very nice work. I hope it will attract a wide audience during the conference.
> > > > Best regards.

---

> > > > > ### Comment · ~Sung_Ju_Hwang1 · 2021-05-04
> > > > > **Thank you**
> > > > >
> > > > > Thank you for the helpful comments. We have discussed Rao et al. 19 in the related work section of the revision, and have also acknowledged your (and all other anonymous reviewers') participation in the discussion, in the acknowledgements. Please check the revised version of the paper when you have time.

---

> > > > > > ### Comment · ~D._Khuê_Lê-Huu1 · 2021-05-04
> > > > > > **I am entirely satisfied with the updates. Thank you!**
> > > > > >
> > > > > > Thank you very much for your updates! The discussion on CURL looks good to me. Thanks also for the kind message in the acknowledgements (it was not needed though, but I appreciate it).
> > > > > > Best regards.

---

### Decision · Program_Chairs · 2021-01-07
**Final Decision**

**Decision:**

Accept (Spotlight)

**Comment:**

This paper addresses a method for unsupervised meta-learning where a VAE with Gaussian mixture prior is used and set-level inference, taking episode-specific dataset as input, is performed to calculate its posterior. In the meta-testing phase, semi-supervised learning with the learned VAE is used to fast adapt to few-show learning. Reviewers are satisfied with the author responses, agreeing that the method is a principled way to tackle unsupervised meta-learning.